# THE EFFECT OF ADVERSARIAL TRAINING: A THEORETICAL CHARACTERIZATION

## ABSTRACT

It has widely shown that adversarial training (Madry et al., 2018) is effective in defending adversarial attack empirically. However, the theoretical understanding of the difference between the solution of adversarial training and that of standard training is limited. In this paper, we characterize the solution of adversarial training for linear classification problem for a full range of adversarial radius $\varepsilon$. Specifically, we show that if the data themselves are "$\varepsilon$-strongly linearly separable", adversarial training with radius smaller than $\varepsilon$ converges to the hard margin solution of SVM with a faster rate than standard training. If the data themselves are not "$\varepsilon$-strongly linearly separable", we show that adversarial training with radius $\varepsilon$ is stable to outliers while standard training is not. Moreover, we prove that the classifier returned by adversarial training with a large radius $\varepsilon$ has low confidence in each data point. Experiments corroborate our theoretical finding well.

## 1 INTRODUCTION

Despite the impressive performance of deep neural networks on various learning tasks, the widely existing adversarial examples (Goodfellow et al., 2014; Szegedy et al., 2017) has thwarted its application in the safety-sensitive scenarios (Kurakin et al., 2016; Chen et al., 2015). A well trained neural network can be vulnerable to certain small adversarial perturbation added to the original data, despite the perturbations is almost imperceptible to human.

Adversarial training (Madry et al., 2018) is an effective method to train a robust deep neural network that can resist adversarial samples to some extent. However, the theoretical understanding of adversarial training is quite limited. In this paper, we try to unveil the mystery of adversarial training. Specifically, in this paper for adversarial training, we consider the attack in $l_2$ norm, i.e. the following objective

$$\min_{\boldsymbol{\theta}} \frac{1}{N} \sum_{i=1}^{N} \max_{\|\boldsymbol{x} - \boldsymbol{x}_i\|_2 \leq \varepsilon} \ell(\boldsymbol{x}, \boldsymbol{\theta}). \tag{1}$$

This is in contrast with the standard training objective $\min_{\boldsymbol{\theta}} \frac{1}{N} \sum_{i=1}^{N} \ell(\boldsymbol{x}, \boldsymbol{\theta})$. To obtain a clear theoretical characterization, we focus on the linear classifier.

In fact, for linearly separable data, Soudry et al. (2017); Ji & Telgarsky (2018) have proven that the linear classifier trained by gradient descent (GD) with logistic loss converges to the hard margin solution of SVM with a rate of $O((\log t)^{-1})$. However, we find things become much different for adversarial training. We first prove that for "$\varepsilon$-strongly linearly separable" data (Definition 1), GD can find the hard margin classifier with a faster rate of $O((\log t)^{-(1+\varepsilon^*)})$, with the same exponential tail loss used in Soudry et al. (2017). Here $\varepsilon^* = (|\mathbf{S}|\varepsilon(\varepsilon_1 - \varepsilon))/N$, where $\varepsilon_1$ is the distance between the support vectors and the hard margin solution of SVM, and $|\mathbf{S}|$ is the number of support vectors corresponding to hard margin classifier. This result shows that we can find a robust solution much faster by adversarial training if the data themselves are more separable than the adversarial radius.

When the data are not $\varepsilon$-strongly linearly separable, adversarial training gives significantly different solutions from the standard training. To further illustration, we consider the following case, most of the data are $\varepsilon$-strongly linearly separable while there several outliers are not or even not linearly separable. Then the classifier returned by standard training is heavily affected by these outliers, since the classifier returned by standard training converges to the hard margin classifier while it can

be sensitive to outliers. However, we can show the stability of adversarial training to outliers, i.e. the classifier returned by adversarial training is slightly affected by outliers. Next, we also show that adversarial training leads to a classifier with relatively lower confidence in each data point than that of standard training. We then give a formal characterization for this phenomenon under the case of a large $\varepsilon$. The low confidence in each training data naturally induces a high training loss. A simple generalization error bound informs the high loss on test set, which interprets the widely observed poor test performance of adversarial training.

## 1.1 RELATED WORK

Plenty of work trying to obtain a large margin solution to promote model robustness. Cisse et al. (2017); Hein & Andriushchenko (2017) regularize the training with the Lipschitz constant of the model to enhance robustness. Another line of work (Elsayed et al., 2018; Lee et al., 2019a; Serra et al., 2018; Croce et al., 2019) uses a first order approximation to compute the margin of deep neural network, and then find a large margin solution by setting the approximation to be the optimization objective. However, none of these works come up with a theoretical guarantee.

Our paper is also related with Soudry et al. (2017); Ji & Telgarsky (2018) who prove that gradient descent with logistic loss converge to the hard margin classifier for linearly separable data but with a rate of $O((\log t)^{-1})$ or on non-linearly separable data with rate of $O(\log \log t / \log t)$. Nacson et al. (2018) extend this result to stochastic gradient descent. Gunasekar et al. (2018) study the convergence under other optimization methods. Lyu & Li (2019) provides a similar result for the homogeneous neural networks. However, they all target the standard optimization problem rather than the adversarial training.

Ilyas et al. (2019) studies adversarial training via the lens of robust/non-robust features. They claim that adversarial attacks are attributed to the presence of non-robust features and adversarial training can be viewed as explicitly preventing the classifier from learning useful but non-robust features. Their theory is presented under the parameter estimation framework, which is different from the conventional adversarial training. Moreover, they do not discuss the convergence direction of adversarial training.

## 2 NOTATIONS AND ASSUMPTIONS

In this section, we introduce the notations and assumptions we used in this paper. We consider a binary classification problem.

**Notations.** Dataset is represented as $\{\boldsymbol{x}_i, y_i\}_{i=1}^{N}$, where $\boldsymbol{x}_i \in \mathbb{R}^d$ and $y_i \in \{-1, 1\}$. The loss function is represented by $\ell(\cdot)$. $\|\cdot\|$ means $l_2$ norm in this paper. The objective of standard training can be represented as

$$\mathcal{L}_0(\boldsymbol{w}) = \frac{1}{N} \sum_{i=1}^{N} \ell(\boldsymbol{w}^T y_i \boldsymbol{x}_i), \tag{2}$$

where $\boldsymbol{w} \in \mathbb{R}^d$ is a linear classifier. Adversarial training has the objective of

$$\mathcal{L}(\boldsymbol{w}) = \frac{1}{N} \sum_{i=1}^{N} \max_{\|\boldsymbol{x} - \boldsymbol{x}_i\| \le \varepsilon} \ell(\boldsymbol{w}^T y_i \boldsymbol{x}), \tag{3}$$

where $\varepsilon$ is the adversarial radius. An intuitive explanation of adversarial training is that it requires the classifier perform well in the $\varepsilon$-ball centered at each data $\boldsymbol{x}_i$. In this paper, $\boldsymbol{w}_t$ is the iterate at step $t$ of gradient descent (GD) under adversarial training. We use $\hat{\boldsymbol{w}}$ to represent the solution of SVM (8). Now, we give the definition of $\varepsilon$-strongly linearly separable data.

**Definition 1** ($\varepsilon$-strongly linearly separable). [1] *The dataset is $\varepsilon$-strongly linearly separable, if there exists $\boldsymbol{w}^*$ such that $\forall \boldsymbol{x}_i, \boldsymbol{w}^{*T} y_i \boldsymbol{x}_i > 0$ and $\boldsymbol{w}^{*T} y_i \boldsymbol{x}_i > \varepsilon \|\boldsymbol{w}^*\|$.*

The $\varepsilon$-strongly linearly separable means there exists a linear classifier that does not only give a right prediction to each data point but also ensures all data are away from the linear classifier larger than

---

[1]Please notice that $\varepsilon$-strongly linearly separable is a stronger condition compared with linearly separable. It distinguishes with the separability under soft margin which allows some $\boldsymbol{w}^{*T} y_i \boldsymbol{x}_i < 0$ but $\boldsymbol{w}^{*T} y_i \boldsymbol{x}_i > \varepsilon \|\boldsymbol{w}^*\|$.

$\varepsilon$. In addition, linearly separable refers to choosing $\varepsilon = 0$ in Definition 1. We now define confidence of classifier.

**Definition 2** (Confidence). *For a given linear classifier $\boldsymbol{w}$ of binary classification problem, the logits is $1/(1 + e^{-\boldsymbol{w}^T \boldsymbol{x}_i})$ and $e^{-\boldsymbol{w}^T \boldsymbol{x}_i}/(1 + e^{-\boldsymbol{w}^T \boldsymbol{x}_i})$ for category 1 and $-1$. Then $\max\{1/(1 + e^{-\boldsymbol{w}^T \boldsymbol{x}_i}), e^{-\boldsymbol{w}^T \boldsymbol{x}_i}/(1 + e^{-\boldsymbol{w}^T \boldsymbol{x}_i})\}$ is the confidence of $\boldsymbol{w}$ on data $\boldsymbol{x}_i$.*

The confidence measures how confident the classifier is about the prediction. In the sequel, we use $\boldsymbol{x}_i$ to represent $y_i \boldsymbol{x}_i$ for simplify the notations. We introduce the assumptions used in this paper.

**Assumption 1.** *The loss function $\ell(\cdot)$[2] satisfies that $\forall u, \ell(u) > 0, \ell'(u) < 0, l''(u) \geq 0, \lim_{u \to \infty} \ell(u) = \lim_{u \to \infty} \ell'(u) = 0$, and the $\ell(u)$ has Lipschitz gradient.[3]*

**Assumption 2.** *The $l(\cdot)$ and $l'(\cdot)$ have exponential tail, which means there exists some constant $u_0$ and $C_1, C_2$ satisfies that $\forall u > u_0, C_1 e^{-u} \leq l(u) \leq C_2 e^{-u}$ as well as $l'(u)$.[4]*

**Assumption 3.** *The $\boldsymbol{w}^T \boldsymbol{x}_i - \varepsilon \|\boldsymbol{w}\|$ has the range of $[c_1, \infty)$ for each $\boldsymbol{x}_i$, where $c_1 > -\infty$. $\|\boldsymbol{w}\| \geq c_2$ for some constant $c_2$.[5].*

Due to the properties of loss function, the adversarial training objective (3) with radius $\varepsilon$ has an explicit formula

$$\mathcal{L}(\boldsymbol{w}) = \frac{1}{N} \sum_{i=1}^{N} \ell\left(\boldsymbol{w}^T \boldsymbol{x}_i - \varepsilon \|\boldsymbol{w}\|\right). \tag{4}$$

The adversarial training objective equation 4 is optimized by the gradient descent, i.e.,

$$\boldsymbol{w}_{t+1} = \boldsymbol{w}_t - \eta \nabla \mathcal{L}(\boldsymbol{w}_t) = \boldsymbol{w}_t - \eta \sum_{i=1}^{N} \ell'\left(\boldsymbol{w}_t^T \boldsymbol{x}_i - \varepsilon \|\boldsymbol{w}_t\|\right)\left(\boldsymbol{x}_i - \varepsilon \frac{\boldsymbol{w}_t}{\|\boldsymbol{w}_t\|}\right), \tag{5}$$

where $\eta$ is the learning rate.

## 3 ADVERSARIAL TRAINING CONVERGES FASTER TO HARD MARGIN FOR $\varepsilon$-STRONGLY LINEARLY SEPARABLE DATA

In this section, we theoretically characterize where adversarial training converges when the data themselves are $\varepsilon$-strongly linearly separable. We first have the following key lemma which ensures the loss of adversarial training with radius $\varepsilon$ can converge to zero on $\varepsilon$-strongly linearly separable data. The proof of this lemma is delegated to Appendix A.

**Lemma 1.** *The adversarial training objective equation 4 is convex and L-smooth for some positive constant L. Then if the data are $\varepsilon$-strongly linearly separable (1), the iterates $\{\boldsymbol{w}_t\}$ returned by GD under adversarial training satisfy 1): $\lim_{t \to \infty} \mathcal{L}(\boldsymbol{w}_t) = 0$, 2): $\lim_{t \to \infty} \|\boldsymbol{w}_t\| = \infty$. and 3): $\lim_{t \to \infty} \boldsymbol{w}_t^T \boldsymbol{x}_i - \varepsilon \|\boldsymbol{w}_t\| = \infty$*

The distance of $\boldsymbol{x}_i$ away from linear classifier $\boldsymbol{w}_t$ is $|\boldsymbol{w}_t^T \boldsymbol{x}_i|/\|\boldsymbol{w}_t\|$. Then, the convergence result in the Lemma 1 does not inform us the robustness of trained classifier. Fortunately, the results in Soudry et al. (2017); Ji & Telgarsky (2018) claim that the iterates of standard training updated by GD can converge to hard margin classifier with the rate of $O((\log t)^{-1})$. Now, we give the main result of this section. The conclusion is similar to Theorem 3 in Soudry et al. (2017) while the convergence is much sharper.

**Theorem 1.** *For any $\varepsilon$-strongly linearly separable data (Definition 1), and loss function $\ell(\cdot)$ satisfies Assumption 1 and 2. If the Assumption 3 holds, then the gradient flow iterates $\boldsymbol{w}(t)$,*

$$\frac{d\boldsymbol{w}(t)}{dt} = -\nabla \mathcal{L}(\boldsymbol{w}(t)) \tag{6}$$

*satisfies*

$$\boldsymbol{w}(t) = \hat{\boldsymbol{w}} \cdot O\left(\log t\right) + \boldsymbol{h}(t), \tag{7}$$

---

[2]The loss is set to be positive, differentiable, monotonically decreasing convex function. Lots of general used loss functions are satisfied, such as $e^{-u}, \log\left(1 + e^{-u}\right)$ etc.

[3]It means that $|\ell'(u) - \ell'(v)| \leq L|u - v|$ for some positive constant $L$.

[4]Lots of loss functions we used such as $\log\left(1 + e^{-u}\right), e^{-u}$ are satisfied with this assumption.

[5]This assumption can be attained by re-scale the norm of $w$ if necessary.

*for a large $t$. Here $\hat{\boldsymbol{w}}$ is the hard margin solution of SVM:*

$$\hat{\boldsymbol{w}} = \arg\min_{\boldsymbol{w}\in\mathbb{R}^d} \|\boldsymbol{w}\| \qquad \text{s.t. } \boldsymbol{w}^T\boldsymbol{x}_i \geq 1. \tag{8}$$

$\|\boldsymbol{h}(t)\|$ *is in the order of* $o(\log t)$. *Then*

$$\lim_{t\to\infty} \left\| \frac{\boldsymbol{w}(t)}{\|\boldsymbol{w}(t)\|} - \frac{\hat{\boldsymbol{w}}}{\|\hat{\boldsymbol{w}}\|} \right\| \leq O((\log t)^{-(1+\varepsilon^*)}), \tag{9}$$

*where* $\varepsilon^* = \frac{|\mathbf{S}|\varepsilon(\varepsilon_1-\varepsilon)}{N}$, $\{\varepsilon_i\}$ *is the sorted distance of data away from the hard margin solution of SVM, and* $\mathbf{S} = \{i : \hat{\boldsymbol{w}}^T\boldsymbol{x}_i = \varepsilon_1\|\hat{\boldsymbol{w}}\|\}$; $|\mathbf{S}|$ *is the number of elements in set* $\mathbf{S}$. *Finally, iterates* $\{\boldsymbol{w}_t\}$ *of adversarial training updated by GD with step size* $\eta$ *satisfies*

$$\|\boldsymbol{w}_k - \boldsymbol{w}(k\eta)\| \leq O(\eta), \tag{10}$$

*for* $k \in \mathbb{N}^+$. *Then we can conclude*

$$\lim_{t\to\infty} \left\| \frac{\boldsymbol{w}_t}{\|\boldsymbol{w}_t\|} - \frac{\hat{\boldsymbol{w}}}{\|\hat{\boldsymbol{w}}\|} \right\| \leq \lim_{t\to\infty} \left\| \frac{\boldsymbol{w}(t\eta)}{\|\boldsymbol{w}(t\eta)\|} - \frac{\hat{\boldsymbol{w}}}{\|\hat{\boldsymbol{w}}\|} \right\| + O(\eta). \tag{11}$$

We give a brief discussion to hard margin solution of SVM. Since the dataset is $\varepsilon$-strongly linearly separable, by KKT condition, we have $\varepsilon < 1/\|\hat{\boldsymbol{w}}\|$ which informs us all the distance between $\boldsymbol{x}_i$ and $\hat{\boldsymbol{w}}$ are larger than $\varepsilon$. Then, we present that perturbations smaller than $\varepsilon$ can be defended by $\hat{\boldsymbol{w}}$. Hence, the linear classifier $\boldsymbol{w}_t$ becomes robust for a large $t$, since it has the same direction with $\hat{\boldsymbol{w}}$. We then conclude adversarial training on $\varepsilon$-strongly linearly separable data helps iterates $\{\boldsymbol{w}_t\}$ converge to a robust solution with a fast rate. A detailed proof of this Theorem is delegated to Appendix C.

For the theorem, we have several remarks about the adversarial training on $\varepsilon$-strongly linearly separable data.

- Adversarial training converges to a robust solution, the same as standard training, with a faster rate. The acceleration is determined by the constant $\varepsilon^* = (|\mathbf{S}|\varepsilon(\varepsilon_1 - \varepsilon))/N$ which related with the true margin $\varepsilon_1$ and the number of support vectors $|\mathbf{S}|$.
- Large proportion of support vectors $|\mathbf{S}|/N$, and an appropriate choice of $\varepsilon$ ($\varepsilon_1/2$ is the best) increase the convergence rate.

We next study where adversarial training converges when the data are not $\varepsilon$-strongly linearly separable.

## 4 Adversarial Training When Data Are Not $\varepsilon$-strongly linearly separable

It's hard to obtain the exact separability of data without verifying the hard margin solution of SVM before adversarial training. Hence, understanding adversarial training on non-$\varepsilon$-strongly linearly separable data is also crucial.

### 4.1 The Influence of Outliers

We consider the case that clean data are $\varepsilon$-strongly linearly separable but the whole data set is not $\varepsilon$-strongly linearly separable because of outliers. We investigate how the outliers affect standard training and adversarial training.

### 4.1.1 Standard Training Is Unstable to Outliers

We now illustrate how standard training can be easily altered by outliers for both linearly and non-linearly separable data.

As shown in Soudry et al. (2017); Ji & Telgarsky (2018), gradient descent converges to the hard margin solution for the exponential loss and linear separable data. Although the hard margin solution is believed to be a reasonably good classifier, it has potential to be non-robust, if one can intentionally insert outliers to be support vectors. The following proposition presents that hard margin classifier somehow can be non-robust.

**Proposition 1.** *For two linearly separable datasets $\{x_i^1\}_{i=1}^{N_1}$ and $\{x_i^2\}_{i=1}^{N_2}$, let $\hat{w}_1$ and $\hat{w}_2$ be their hard margin classifier respectively. Then, the hard margin classifier $\hat{w}$ of union dataset $\{x_i^1\}_{i=1}^{N_1} \bigcup \{x_i^2\}_{i=1}^{N_2}$ satisfies $\|\hat{w}\| \geq \max\{\|\hat{w}_1\|, \|\hat{w}_2\|\}$.*

This proposition can be derived from the definition of hard margin classifier, which suggests that adding some extra outliers to dataset can easily effect the robustness of hard margin classifier, since $\|\hat{w}\| \geq \max\{\|\hat{w}_1\|, \|\hat{w}_2\|\}$ and one of $w_i$ can be extremely large. Hence, the non-robustness of hard margin classifier can be attributed to some outliers. We use the next example to illustrate it.

**Example 1.** *Suppose a dataset $\left\{\left(x_i^1, \mathbf{0}\right)\right\}_{i=1}^{N_1}$ is well linearly separable with large margin, which means the hard margin classifier $\hat{w}_1$ has a small norm. We insert some outliers which have the form $\left\{\left(\mathbf{0}, x_i^2,\right)\right\}_{i=1}^{N_2}$ and $N_1 \gg N_2$. If the outlier is linearly separable with relatively small margin, the corresponding hard margin solution is $\hat{w}_2$, then we have $\|\hat{w}_1\| \ll \|\hat{w}_2\|$.*

We can show that hard margin classifier on the union dataset $\left\{\left(x_i^1, \mathbf{0}\right)\right\}_{i=1}^{N_1} \bigcup \left\{\left(\mathbf{0}, x_i^2\right)\right\}_{i=1}^{N_2}$ is $(\hat{w}_1, \hat{w}_2)$. Then for each $\left(x_i^1, \mathbf{0}\right)$, we have

$$\frac{\left|(\hat{w}_1, \hat{w}_2)^T \left(x_i^1, \mathbf{0}\right)\right|}{\|(\hat{w}_1, \hat{w}_2)\|} = \frac{|\hat{w}_1^T x_i^1|}{\sqrt{\|\hat{w}_1\|^2 + \|\hat{w}_2\|^2}} \leq \frac{|\hat{w}_1^T x_i^1|}{\|\hat{w}_2\|} \ll \frac{|\hat{w}_1^T x_i^1|}{\|\hat{w}_1\|}. \tag{12}$$

This shows the distance between each $\left\{\left(x_i^1, \mathbf{0}\right)\right\}_{i=1}^{N_1}$ and new hard margin classifier $(\hat{w}_1, \hat{w}_2)$ become extremely small due to a few outliers $\left\{\left(\mathbf{0}, x_i^2\right)\right\}_{i=1}^{N_2}$.

This indicates the solution returned by standard training is somewhat very sensitive to a few outliers because outliers can easily alter the hard margin classifier.

We now give a more general description of outliers based on the understanding of Example 1. Given two datasets $\{x_i^1\}_{i=1}^{N_1}$ and $\{x_i^2\}_{i=1}^{N_2}$ with $N_1, N_2$ data respectively, where $N_1 \gg N_2$. $\{x_i^1\}_{i=1}^{N_1}$ themselves are $\varepsilon$-strongly linearly separable , but $\{x_i^2\}_{i=1}^{N_2}$ are not. Then, dataset $\{x_i^2\}_{i=1}^{N_2}$ is considered as outliers. Outliers themselves can even be not linearly separable. We use the next example to illustrate another behavior of standard training to non-linearly separable outliers.

**Example 2.** *Dataset $\{(x_i^{11}, x_i^{12})\}_{i=1}^{N_1} \in \mathbb{R}^{d_1+d_2}$ is $\varepsilon$-strongly linearly separable but $\{(\mathbf{0}, x_i^{22})\}_{i=1}^{N_2} \in \mathbb{R}^{d_1+d_2}$ is not linearly separable. Let $S = \mathrm{span}\{x_1^{12}, \cdots, x_{N_1}^{12}\} = \mathrm{span}\{x_1^{22}, \cdots, x_{N_2}^{22}\}$. Without loss of generality, we assume $\{x_i^{11}\}_{i=1}^{N_1}$ themselves are linearly separable but the hard margin classifier of them $\hat{w}_{11}$ has relatively small margin compared to the $\{x_i^{12}\}_{i=1}^{N_1}$, i.e., $\|\hat{w}_{11}\| \gg \|\hat{w}_{12}\|$, where $\hat{w}_{12}$ is the hard margin classifier on $\{x_i^{12}\}_{i=1}^{N_1}$ [6].*

The union datasets $\{(x_i^{11}, x_i^{12})\}_{i=1}^{N_1} \bigcup \{(\mathbf{0}, x_i^{22})\}_{i=1}^{N_2}$ is not linearly separable because of the outliers set $\{(\mathbf{0}, x_i^{22})\}_{i=1}^{N_2}$. Even worse, we show in this case, the classifier returned by standard training can make the prediction most based on the useful but non-robust information $\{x_i^{11}\}_{i=1}^{N_1}$ (Ilyas et al., 2019).

The reason is as follows. The outliers $\{(\mathbf{0}, x_i^{22})\}_{i=1}^{N_2}$ happens to be the "Strongly Convex Part" as in Theorem 4.1 of Ji & Telgarsky (2018), then the iterates $\{w_t\}$ updated by gradient descent of standard training converge to a direction of the hard margin classifier $\hat{w}_{11}$ of dataset $\{\Pi_{S^\perp}(x_i^{11}, x_i^{12})\}_{i=1}^{N_1} = \{(x_i^{11}, \mathbf{0})\}_{i=1}^{N_1}$, where $\Pi_{S^\perp}(\cdot)$ is the projection operator of space $S^\perp$. Then, the presence of outliers $\{(\mathbf{0}, x_i^{22})\}_{i=1}^{N_2}$ renders the classifier returned by standard training on union datasets with poor robustness while the true hard margin classifier on dataset $\{(x_i^{11}, x_i^{12})\}_{i=1}^{N_1}$ is robust.

The two examples that standard training converges to non-robust solution at the presence of outliers also demonstrates that standard training can easily capture the "non-robust but useful information". The first is to classify outliers $\{(\mathbf{0}, x_i^{22})\}_{i=1}^{N_2}$ into correct category renders the classifier $(\hat{w}_1, \hat{w}_2)$ to be non-robust. The second describes that the outliers mute the information from $\{(\mathbf{0}, x_i^{12})\}_{i=1}^{N_1}$ for dataset $\{(x_i^{11}, x_i^{12})\}_{i=1}^{N_1}$. Then, the classifier turns into a non-robust but useful classifiers.

---

[6] The hard margin classifier of $\{(x_i^{11}, x_i^{12})\}_{i=1}^{N_1}$ $\hat{w}_1$ satisfies $\|\hat{w}_1\| \leq \min\{\|\hat{w}_{11}\|, \|\hat{w}_{12}\|\}$ which leads to a even more robust solution.

### 4.1.2 Adversarial Training is Stable to Outliers

Now, we turn to the behavior of adversarial training on the above two examples. Lemma 1 shows that the iterates $\{\boldsymbol{w}_t\}$ of adversarial training on $\varepsilon$-strongly linearly separable data updated by GD can satisfy $\lim_{t\to\infty}\|\boldsymbol{w}_t\| = \infty$. But the non-$\varepsilon$-strongly linearly separability of data ensures there exists some $\boldsymbol{x}_i^k$ such that $\boldsymbol{w}_t^T\boldsymbol{x}_i^k - \varepsilon\|\boldsymbol{w}_t\| \leq 0$ for each specific $t$. Hence, $\|\boldsymbol{w}_t\|$ can not go to infinity, otherwise we end up with an infinite loss. This is the essential distinction caused by non-$\varepsilon$-strongly linearly separability. Now, we give a formal analysis to the stability of adversarial training to outliers.

Due to Lemma 1, iterates of adversarial training updated by GD converge to a minimum $\boldsymbol{w}^*$. Then, let $l'\left(\boldsymbol{w}^{*T}\boldsymbol{x}_i^k - \varepsilon\|\boldsymbol{w}^*\|\right) = p_i^k$, we have

$$\nabla\mathcal{L}(\boldsymbol{w}^*) = \frac{1}{N_1 + N_2}\sum_{k=1}^{2}\sum_{i=1}^{N_k} p_i^k\left(\boldsymbol{x}_i^k - \varepsilon\frac{\boldsymbol{w}^*}{\|\boldsymbol{w}^*\|}\right) = 0. \tag{13}$$

It gives

$$\varepsilon\frac{\boldsymbol{w}^*}{\|\boldsymbol{w}^*\|}\sum_{k=1}^{2}\sum_{i=1}^{N_k} p_i^k = \sum_{k=1}^{2}\sum_{i=1}^{N_k} p_i^k\boldsymbol{x}_i^k = \sum_{k=1}^{2}\mathbf{X}_k\boldsymbol{p}_k, \tag{14}$$

where $\mathbf{X}_k = (\boldsymbol{x}_1^k, \cdots, \boldsymbol{x}_{N_k}^k)$ and $\boldsymbol{p}_k = (p_1^k, \cdots p_{N_k}^k)$. This implies that the direction of vector $\boldsymbol{w}^*$ is decided by a linear combination of data points. $\mathbf{X}_k\boldsymbol{p}_k$ represents the contribution brought by $\{\boldsymbol{x}_i^k\}_{i=1}^{N_k}$.

When data are $\varepsilon$-strongly linearly separable, by Assumption 2, we see $p_i^k/p_j^{k'} = o(1)$ ($\|\boldsymbol{w}_t\|$ goes to infinity) for any support vector $\boldsymbol{x}_i^k$ and non-support vector $\boldsymbol{x}_j^{k'}$ of $\boldsymbol{w}^*/\|\boldsymbol{w}^*\|$. Hence, the direction of $\boldsymbol{w}^*$ is mostly decided by support vectors. This is consistent with the conclusion that hard margin solution of SVM $\hat{\boldsymbol{w}}$ satisfies (KKT condition) $\hat{\boldsymbol{w}} = \sum_{k=1}^{2}\sum_{i=1}^{N_k}\alpha_i^k\boldsymbol{x}_i^k$, where $\alpha_i^k = 0$ for non-support vectors. Then, hard margin classifier is only decided by the linear combination of support vectors, which could be potential instability of standard training to outliers, since outliers can easily alter support vectors.

However, things become different for non-$\varepsilon$-strongly linearly separable data. Since $\|\boldsymbol{w}_t\|$ can not go to infinity, we have the relation $p_i^k/p_j^{k'} = O(1)$. Then the following inequality

$$\frac{\lambda_{\mathbf{min}}^{\frac{1}{2}}\left(\mathbf{X}_2^T\mathbf{X}_2\right)\|\boldsymbol{p}_2\|}{\lambda_{\mathbf{max}}^{\frac{1}{2}}\left(\mathbf{X}_1^T\mathbf{X}_1\right)\|\boldsymbol{p}_1\|} \leq \frac{\|\mathbf{X}_2\boldsymbol{p}_2\|}{\|\mathbf{X}_1\boldsymbol{p}_1\|} \leq \frac{\lambda_{\mathbf{max}}^{\frac{1}{2}}\left(\mathbf{X}_2^T\mathbf{X}_2\right)\|\boldsymbol{p}_2\|}{\lambda_{\mathbf{min}}^{\frac{1}{2}}\left(\mathbf{X}_1^T\mathbf{X}_1\right)\|\boldsymbol{p}_1\|}, \tag{15}$$

informs us the stability of adversarial training to outliers, where $\lambda_{\mathbf{min}}(\mathbf{A})$, $\lambda_{\mathbf{max}}(\mathbf{A})$ are respectively the smallest and largest eigenvalue of matrix $\mathbf{A}$. From equation 13, we see the direction of $\boldsymbol{w}^*$ can be decided by both dataset $\{\boldsymbol{x}_i^1\}_{i=1}^{N_1}$ and $\{\boldsymbol{x}_i^2\}_{i=1}^{N_2}$. Since $p_i^1$ has the same scale with $p_j^2$ for every $i, j$ and $N_1 \gg N_2$, we see $\|\boldsymbol{p}_1\| \gg \|\boldsymbol{p}_2\|$, then $\|\mathbf{X}_1\boldsymbol{p}_1\| \gg \|\mathbf{X}_2\boldsymbol{p}_2\|$. Hence, equation 15 implies that $\mathbf{X}_2\boldsymbol{p}_2$ has little contribution to the direction of $\boldsymbol{w}^*$. Thus, $\boldsymbol{w}^*$ can not be the hard margin classifier on union dataset. In addition, $\boldsymbol{w}^*$ is mostly decided by data $\{\boldsymbol{x}_i^1\}_{i=1}^{N_1}$ which tells the stability of adversarial training to outliers.

We now provide a discussion to the stability of adversarial training via the lens of robust/non-robust but useful features proposed by Ilyas et al. (2019). First, we quote the definition of non-robust but useful feature here.

**Definition 3** (non-robust but useful feature (Ilyas et al., 2019))**.** *For a given distribution $(\boldsymbol{x}, y)$, if $\mathbb{E}_{(\boldsymbol{x},y)}[y \cdot f(\boldsymbol{x})] \geq \rho$ for some $\rho > 0$, but $\mathbb{E}_{(\boldsymbol{x},y)}\left[\inf_{\|\boldsymbol{\delta}\|\leq\varepsilon} y \cdot f(\boldsymbol{x} + \boldsymbol{\delta})\right] \leq 0$, then $f(\cdot)$ is non-robust but useful feature.*

The definition of robust feature can be generalized for $f(\cdot)$ with $\mathbb{E}_{(\boldsymbol{x},y)}\left[\inf_{\|\boldsymbol{\delta}\|\leq\varepsilon} y \cdot f(\boldsymbol{x} + \boldsymbol{\delta})\right] \geq 0$. We see the feature is defined by classifier $f(\cdot)$ which is $\boldsymbol{w}$ for linear case. Then, the hard margin solution of SVM $\hat{\boldsymbol{w}}_1, \hat{\boldsymbol{w}}_2$ for $\{\boldsymbol{x}_i^1\}_{i=1}^{N_1}$ and $\{\boldsymbol{x}_i^2\}_{i=1}^{N_2}$ in Example 1 and 2 are respectively robust feature and non-robust feature. Further more, let $\boldsymbol{w}^*$ be the classifier (feature) captured by adversarial training. We note that $\boldsymbol{w}^*$ being stable to outliers means adversarial training can prevent the classifier capture information from non-robust but useful features in sharp contrast with standard training. This interprets the conclusion in Ilyas et al. (2019) while they only empirically verify it.

We can also understand other scenarios from equation 15 that are not limited to $N_1 \gg N_2$. If $N_1 \approx N_2$, we see adversarial training can preserve a part of information from robust features. Moreover, $N_2 \gg N_1$ means that most features are non-robust, and then adversarial training is also helpless.

## 4.2 CONFIDENCE OF ADVERSARIAL TRAINING

In this subsection, we give a characterization to the confidence of classifier returned by adversarial training where data is not $\varepsilon$-strongly linearly separable. As shown in Section 4.1.2, $\|w_t\|$ converges to some constant when data is not $\varepsilon$-strongly linearly separable. According to Definition 2, the confidence of classifier is decided by $|w_t^T x_i|$ on each $x_i$. Then, a bounded $\|w_t\|$ corresponds to a relatively low confidence in each $x_i$.

To give a formal description to the confidence of $w_t$, we discuss an extremely large $\varepsilon \geq \frac{1}{2} \max_{i,j} \|x_i - x_j\|$, which means each $\varepsilon$-ball centered at $x_i$ has overlap with each other. For a $x_i$, *larger $|w^{*T} x_i|$ corresponds with higher prediction confidence*. For a given dataset $\{x_i\}_{i=1}^N$, let $N_1$, $N_2$ respectively be the number of data in each category and $x_{k,i}$ represents the data from the $k$-th category. Then we have following theorem describing the confidence of classifier.

**Theorem 2.** *Let $l(u) = e^{-u}$, for classifier $w^*$ returned by adversarial training on non-linearly separable data, $|w^{*T} x_{k,i}|$ is either smaller than $\varepsilon \|w^*\|$ or satisfies*

$$e^{\left(|w^{*T} x_{k,i}| - \varepsilon \|w^*\|\right)} \leq \frac{1}{N_{k'}} \sum_{j=1}^{N_{k'}} \max_{x:\|x - x_{k',j}\| \leq \varepsilon} e^{\left(-w^{*T} x + \varepsilon \|w^*\|\right)}, \qquad if \ w^{*T} x_{k,i} \geq \varepsilon \|w^*\|,$$

$$e^{\left(|w^{*T} x_{k,i}|\right)} \leq \frac{1}{N_{k'}} \sum_{j=1}^{N_{k'}} e^{\left(-w^{*T} x_{k',j} - \varepsilon \|w^*\|\right)}, \qquad if \ w^{*T} x_{k,i} \leq -\varepsilon \|w^*\|, w^{*T} x_{k',j} \geq 0,$$

(16)

*for $k' \neq k$.*

Our results also applies to $l(\cdot)$ discussed in this paper. A more explicit upper bound for those $w^{*T} x_{k,i} \geq \varepsilon \|w^*\|$ is referred to equation 56. We notice that $\|w^*\|$ is bounded on non-$\varepsilon$-strongly linearly separable data. Then, this theorem implies that $|w^{*T} x_i|$ on each $x_i$ can not be extremely large. Hence, a relatively low confidence in each $x_i$. A detailed proof of this Theorem is delegated to Appendix D. Here we use $\varepsilon \geq \frac{1}{2} \max_{i,j} \|x_i - x_j\|$ is out of simplicity, a large $\varepsilon$ for adversarial training can also correspond to a similar result.

Theorem 2 informs the training loss of classifier $w^*$ is at a relatively high level. Then, a simple generalization error bound (Vapnik, 1995) can tell that $w^*$ can also end up with a high error on test data. This is an interpretation to the model trained by adversarial training always face a poor test accuracy (Madry et al., 2018).

## 5 EMPIRICALLY STUDY

In this section, we compare adversarial training with standard training via linear classifier for two scenarios, i.e., $\varepsilon$-strongly linearly separable data and non-$\varepsilon$-strongly linearly separable data. All experiments are conducted with loss function $\ell(u) = \log(1 + e^{-u})$, and the update rule is GD with learning rate 0.01.

### 5.1 EXPERIMENTS ON $\varepsilon$-STRONGLY LINEARLY SEPARABLE DATA

We first conduct a series of simulations to verify our conclusion on $\varepsilon$-strongly linearly separable data. The dataset includes support and non-support vectors. We generate non-support vectors $\{x_i\}_{i=1}^N \sim \mathcal{N}(y_i \cdot 2 \cdot \mathbf{1}_2, 0.3 \cdot \mathbf{I}_2)$ and support vectors on the $\mathbf{1}_2^T x_i = y_i \cdot 2$. If there exists at least one support vector in each category, the hard margin solution of SVM $\hat{w}$ is $\mathbf{1}_2$. See Figure 1 for an example. The distance of support vectors away from the hard margin classifier is $\sqrt{2}$. Hence, the data are at most $\sqrt{2}$-linearly separable. We respectively verify our conclusions in Theorem 1 of the convergence rate, support vector number $|\mathbf{S}|$ and adversarial radius $\varepsilon$.

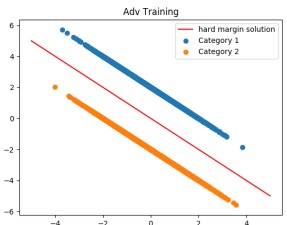 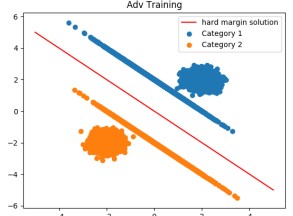 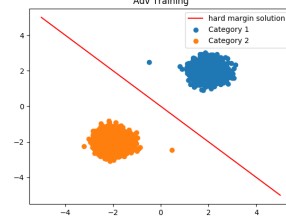

(a) Dataset with 5000 support vectors in each category. (b) Dataset with 2500 support vectors and 2500 non-support vectors in each category. (c) Dataset with 1 support vector and 4999 non-support vectors in each category.

Figure 1: Three types of datasets.

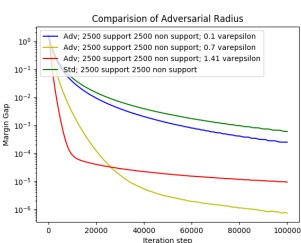 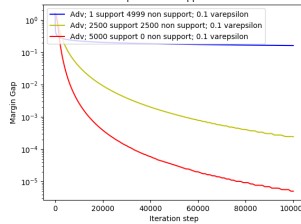 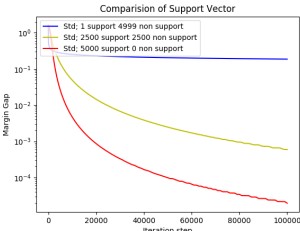

(a) Adversarial training with different $\varepsilon$. (b) Adversarial training with different number of support vectors. (c) Standard training with different number of support vectors

Figure 2: Gap between the direction of iterates and direction of hard margin classifier.

We conduct adversarial training with different $\varepsilon$ and standard training on three datasets: number of support vectors and non-support vectors in each category are respectively $(5000, 0); (2500, 2500)$ and $(1, 4999)$ (see Figure 1). The results can be referred to Figure 2. Some extra experiments are delegated to Appendix E. We should highlight the direction of $\{\boldsymbol{w}_t\}$ here can converge with rate of $O((\log 0.01 \cdot t)^{-1})$ and $O((\log 0.01 \cdot t)^{-(1+\varepsilon^*)})$, respectively for standard training and adversarial training due to equation 11.

The "margin gap" in Figure 2 means distance between the direction of iterates $\boldsymbol{w}_t/\|\boldsymbol{w}_t\|$ and hard margin solution of SVM $\mathbf{1}_2/\sqrt{2}$. In Figure 2, the label, for example "Adv: 2500 support 2500 non support: 0.1 varepsilon" means adversarial training with $\varepsilon = 0.1$ on dataset with 2500 support vectors and 2500 non-support vectors in each category. From the results, we see that adversarial training indeed accelerates convergence rate of $\boldsymbol{w}_t/\|\boldsymbol{w}_t\|$. On the other hand, both adversarial training and standard training can benefit from the number of support vectors. Finally, as we claimed in Theorem 1, the best adversarial radius $\varepsilon$ is $\varepsilon_1/2$ which is $\sqrt{2}/2$ here.

## 5.2 Experiments on Non-$\varepsilon$-Strongly linearly separable Data

In this subsection, we conduct experiments on linearly separable (but non-$\varepsilon$-strongly linearly separable) and non-linearly separable data. More extra experiments can be referred to Appendix E.

We first discuss the stability of adversarial training to outliers on linearly separable data corresponding to Example 1. We generate $\{(\boldsymbol{x}_i^1, \mathbf{0}_2)\}_{i=1}^{10000} \sim (\mathcal{N}(y_i \cdot 2 \cdot \mathbf{1}_2, 0.3 \cdot \mathbf{I}_2), \mathbf{0}_2)$ and $\{(\mathbf{0}_2, \boldsymbol{x}_i^2)\}_{i=1}^{100} \sim (\mathbf{0}_2, \mathcal{N}(y_i \cdot 0.5 \cdot \mathbf{1}_2, 0.1 \cdot \mathbf{I}_2))$. The number of the two datasets in each category are 5000 and 50. The dataset $\{(\mathbf{0}_2, \boldsymbol{x}_i^2)\}_{i=1}^{100}$ is linearly separable while non-$\varepsilon$-strongly linearly separable for $\varepsilon = 1.5$. $\{(\boldsymbol{x}_i^1, \mathbf{0}_2)\}_{i=1}^{10000}$ is $\varepsilon$-strongly linearly separable for $\varepsilon = 1.5$. $\{(\mathbf{0}_2, \boldsymbol{x}_i^2)\}_{i=1}^{100}$ can be viewed as outliers. Then, we compare the stability of adversarial training and standard training conducted on the union dataset to outliers. The adversarial radius $\varepsilon$ is 1.5. Detailed results can be referred to Figure 3a.

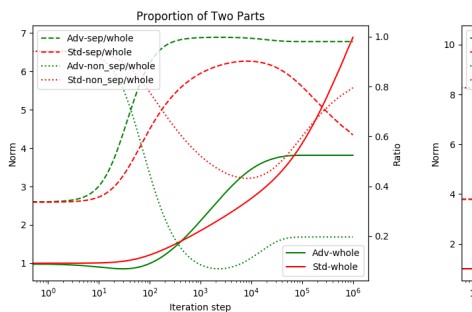 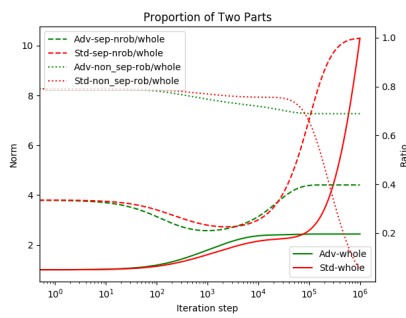

(a) Non-$\varepsilon$-strongly linearly separable but linearly separable data.

(b) Non-linearly separable data.

Figure 3: Adversarial and standard training on non-$\varepsilon$-strongly linearly separable but linearly separable data and non-linearly separable data.

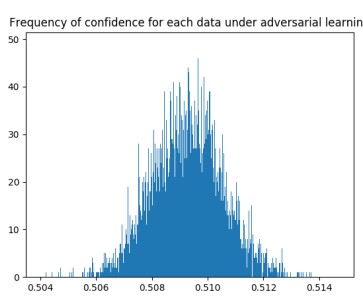 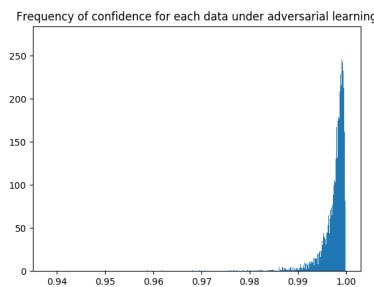

(a) Frequency of confidence for data point under adversarial training

(b) Frequency of confidence for data point under standard learning

Figure 4: The confidence distributions of adversarial training and standard training when data are linearly separable but $\varepsilon \geq \frac{1}{2} \max_{i,j} \|\boldsymbol{x}_i - \boldsymbol{x}_j\|$. The $y$-axis is number of data.

For linearly separable data, the solid lines in Figure 3a are the norm of iterates returned by adversarial training and standard training. The dash lines in Figure 3a, for example, "Adv-sep/whole" means the ratio between the norm of classifier decided by the $\varepsilon$-strongly linearly separable part that is the first two dimensions of iterates and whole norm of iterates ($\|\boldsymbol{w}_1\|/\|\boldsymbol{w}\|$, where $\boldsymbol{w} \in \mathbb{R}^4 = (\boldsymbol{w}_1, \boldsymbol{w}_2) \in \mathbb{R}^{2+2}$). The first and the last two dimensions of iterates are with respect to robust and non-robust features. We have two observations for adversarial training, first, the norm of iterates can converge to some constant rather than infinity. Second, The direction of iterates is mostly decided by $\{(\boldsymbol{x}_i^1, \boldsymbol{0}_2)\}_{i=1}^{10000}$, since the "Adv-sep/whole" can converge to 1 as the iterate steps growing up. However, we have an opposite observation for standard training, which means the direction of classifier obtained by standard training is easily effected by some outliers $\{(\boldsymbol{0}_2, \boldsymbol{x}_i^2)\}_{i=1}^{100}$.

Then, we present the results on non-linearly separable data with respect to Example 2, we generate 5000 $\{(\boldsymbol{x}_i^{11}, \boldsymbol{x}_i^{12})\}_{i=1}^{10000} \sim (\mathcal{N}(y_i \cdot 0.5 \cdot \mathbf{1}_2, 0.1 \cdot \mathbf{I}_2), \mathcal{N}(y_i \cdot 2 \cdot \mathbf{1}_2, 0.3 \cdot \mathbf{I}_2))$ and 50 $\{(\boldsymbol{0}_2, \boldsymbol{x}_i^2)\}_{i=1}^{100} \sim (\boldsymbol{0}_2, \mathcal{N}(-y_i \cdot 0.5 \cdot \mathbf{1}_2, 0.5 \cdot \mathbf{I}_2))$ in each category. Here $\{(\boldsymbol{0}_2, \boldsymbol{x}_i^2)\}_{i=1}^{100}$ is the source of non-linearly separability. $\{\boldsymbol{x}_i^{12}\}_{i=1}^{10000}$ corresponds with a robust hard margin classifier while $\{\boldsymbol{x}_i^{11}\}_{i=1}^{10000}$ is not. $\{(\boldsymbol{0}_2, \boldsymbol{x}_i^2)\}_{i=1}^{100}$ can be viewed as outliers. We conduct adversarial training with $\varepsilon = 0.7$. The results can be referred to Figure 3.

Figure 3b refers to non-linearly separable data, where the solid lines in it are also the norm of iterates. The first and the last two dimensions of iterates are respectively decided by the "useful but non-robust" part and "robust" part. The non-linearly separability is from the last two dimensions of data brought by outliers $\{(\boldsymbol{0}_2, \boldsymbol{x}_i^2)\}_{i=1}^{100}$. The dash lines in Figure 3b, for example, "Adv:sep-nrob" and "Adv-non_sep-rob" are respectively the proportion in norm for the norm of the first and the last two dimensions of iterates ($\|\boldsymbol{w}_1\|/\|\boldsymbol{w}\|$ and $\|\boldsymbol{w}_2\|/\|\boldsymbol{w}\|$, where $\boldsymbol{w} \in \mathbb{R}^4 = (\boldsymbol{w}_1, \boldsymbol{w}_2) \in \mathbb{R}^{2+2}$).

We see standard training can easily capture the information from "useful but non-robust" (first two dimensions) part while the information from the "robust" part (last two dimensions) is muted attributes to outliers $\{(\mathbf{0}_2, \boldsymbol{x}_i^2)\}_{i=1}^{100}$. But adversarial training can oppositely capture the information from "mostly robust" part.

Finally, we see the confidence of the classifier obtained by adversarial training with $\varepsilon \geq \frac{1}{2}\max_{i,j}\|\boldsymbol{x}_i - \boldsymbol{x}_j\|$. We generate $5000$ samples $\{\boldsymbol{x}_i\}_{i=1}^{10000} \sim \mathcal{N}(\pm 0.5 \cdot \mathbf{1}_2, 0.1 \cdot \mathbf{I}_2)$ in each category. The data are linearly separable but non-$\varepsilon$-strongly linearly separable for $\varepsilon = 1$, besides that, $1 \geq \frac{1}{2}\max_{i,j}\|\boldsymbol{x}_i - \boldsymbol{x}_j\|$. We compare the confidence in each data point for classifiers returned by adversarial training with $\varepsilon = 1$ and standard training. The distributions of confidence via classifiers in data can be referred to Figure 4. We see the classifier obtained by adversarial training corresponds with nearly $50\%$ confidence in all data. However, the confidence of classifier returned by standard training is almost $100\%$ in all data. Hence, a large $\varepsilon$ for adversarial training can hurt the prediction confidence of classifier.

## 6 CONCLUSION

In this paper, we give a theoretical characterization to adversarial training for linear classifiers under various settings. We conclude that on $\varepsilon$-strongly linearly separable data, adversarial training helps iterates converge to the hard margin classifier with a rapid rate compared with standard training. It means iterates of adversarial training can be robust with less update steps. Furthermore, we characterize the adversarial training on non-$\varepsilon$-strongly linearly separable data. We show both theoretically and empirically that adversarial training can be more stable to the outliers of the dataset but standard training is not. Finally, we discuss the confidence of the classifier obtained by adversarial training. We prove that under the condition of $\varepsilon \geq \frac{1}{2}\max_{i,j}\|\boldsymbol{x}_i - \boldsymbol{x}_j\|$, the confidence of classifier obtained by adversarial training keeps in a low level. This reveals that a large $\varepsilon$ for adversarial training is not a wise choice.

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

# A   PROOF OF LEMMA 1

*Proof.* We first give convexity to the adversarial training objective. For any two $\boldsymbol{w}_1, \boldsymbol{w}_2$, and $0 \leq \lambda \leq 1$, by the convexity and monotone decreasing property of $\ell(\cdot)$, we have

$$
\frac{1}{N} \sum_{i=1}^{N} \ell \left( \lambda \boldsymbol{w}_1^T \boldsymbol{x}_i + (1-\lambda) \boldsymbol{w}_2^T \boldsymbol{x}_i - \varepsilon \| \lambda \boldsymbol{w}_1 + (1-\lambda) \boldsymbol{w}_2 \| \right)
$$

$$
\leq \frac{1}{N} \sum_{i=1}^{N} \ell \left( \lambda \boldsymbol{w}_1^T \boldsymbol{x}_i + (1-\lambda) \boldsymbol{w}_2^T \boldsymbol{x}_i - \varepsilon \lambda \| \boldsymbol{w}_1 \| - (1-\lambda) \| \boldsymbol{w}_2 \| \right) \tag{17}
$$

$$
\leq \frac{1}{N} \sum_{i=1}^{N} \lambda \ell \left( \boldsymbol{w}_1^T \boldsymbol{x}_i - \varepsilon \| \boldsymbol{w}_1 \| \right) + (1-\lambda) \ell \left( \boldsymbol{w}_2^T \boldsymbol{x}_i - \varepsilon \| \boldsymbol{w}_2 \| \right),
$$

which implies the convexity of adversarial training objective. On the other hand, if $\boldsymbol{u}^T \nabla^2 \mathcal{L}(\boldsymbol{w}) \boldsymbol{u}$ can be bounded by $L \| \boldsymbol{u} \|^2$ for some $L$, then $\mathcal{L}(\boldsymbol{w})$ is $L$-smooth. We see that

$$
\nabla^2 \mathcal{L}(\boldsymbol{w}) = \frac{1}{N} \sum_{i=1}^{N} \ell''(\boldsymbol{w}^T \boldsymbol{x}_i - \varepsilon \| \boldsymbol{w} \|) \left( \boldsymbol{x}_i - \varepsilon \frac{\boldsymbol{w}}{\| \boldsymbol{w} \|} \right) \left( \boldsymbol{x}_i - \varepsilon \frac{\boldsymbol{w}}{\| \boldsymbol{w} \|} \right)^T
$$

$$
- \frac{\varepsilon}{\| \boldsymbol{w} \|^3} \ell'(\boldsymbol{w}^T \boldsymbol{x}_i - \varepsilon \| \boldsymbol{w} \|)(\| \boldsymbol{w} \|^2 \mathbb{I} - \boldsymbol{w} \boldsymbol{w}^T). \tag{18}
$$

Then, $\boldsymbol{u}^T \nabla^2 \mathcal{L}(\boldsymbol{w}) \boldsymbol{u}$ can be divided into two parts, we respectively compute them

$$
\left| \frac{1}{N} \sum_{i=1}^{N} l''(\boldsymbol{w}^T \boldsymbol{x}_i - \varepsilon \| \boldsymbol{w} \|) \left( (\boldsymbol{u}^T \boldsymbol{x}_i)^2 - 2 \frac{\varepsilon}{\| \boldsymbol{w} \|} \boldsymbol{u}^T \boldsymbol{x}_i \boldsymbol{w}^T \boldsymbol{u} + \frac{\varepsilon^2}{\| \boldsymbol{w} \|^2} (\boldsymbol{u}^T \boldsymbol{w})^2 \right) \right|
$$

$$
\leq \frac{L_0}{N} \| \boldsymbol{u} \|^2 \left( \lambda_{\max} \left( \mathbf{X}^T \mathbf{X} \right) + 2 \varepsilon \sum_{i=1}^{N} \| \boldsymbol{x}_i \| + 2 \varepsilon^2 \right). \tag{19}
$$

Here $\mathbf{X}$ is matrix $(\boldsymbol{x}_1^T, \cdots, \boldsymbol{x}_N^T)^T$ and $\lambda_{\max} \left( \mathbf{X}^T \mathbf{X} \right)$ is the largest eigenvalue of $\mathbf{X}^T \mathbf{X}$. On the other hand, according to Assumption 3,

$$
\boldsymbol{u}^T \frac{\varepsilon}{\| \boldsymbol{w} \|^3} \ell'(\boldsymbol{w}^T \boldsymbol{x}_i - \varepsilon \| \boldsymbol{w} \|)(\| \boldsymbol{w} \|^2 \mathbb{I} - \boldsymbol{w} \boldsymbol{w}^T) \boldsymbol{u} \leq 2 \varepsilon \frac{C_1}{c_2} \| \boldsymbol{u} \|^2. \tag{20}
$$

Combining the two above equations, we conclude that largest eigenvalue of $\nabla^2 \mathcal{L}(\boldsymbol{w})$ is bounded for some constant, which result in the $L$-smoothness of adversarial training objective. It's a well known (Boyd & Vandenberghe, 2004) that for a convex and $L$-smooth function $f(\cdot)$, GD with step size $\eta = \frac{1}{L}$ will ensure the iterates $\{x_t\}$ satisfies $f(x_t) - f(x_*) \leq O \left( \frac{1}{t} \right)$. Since the adversarial training loss goes to zero only if $\boldsymbol{w}_t^T \boldsymbol{x}_i - \varepsilon \| \boldsymbol{w}_t \|$ goes to infinity for each $\boldsymbol{x}_i$. It reveals our two conclusions $\lim_{t \to \infty} \| \boldsymbol{w}_t \| = \infty$ and $\lim_{t \to \infty} \boldsymbol{w}_t^T \boldsymbol{x}_i - \varepsilon \| \boldsymbol{w}_t \| = \infty$. □

# B   ORDER OF NORM

In this section we give the order of $\| \boldsymbol{w}(t) \|$ on $\varepsilon$-strongly linearly separable data, which is highly related to the convergence rate of $\left\| \frac{\boldsymbol{w}(t)}{\| \boldsymbol{w}(t) \|} - \frac{\hat{\boldsymbol{w}}}{\| \hat{\boldsymbol{w}} \|} \right\|$. Our proof is based on the gradient flow, we first bound the difference between flow iterates $\boldsymbol{w}(t)$ and $\boldsymbol{w}_t$.

**Lemma 2.** *Let $\boldsymbol{w}_t$ be the iterates updated by GD equation 5, then for $\frac{d\boldsymbol{w}(t)}{dt} = -\nabla \mathcal{L}(\boldsymbol{w}(t))$, we have*

$$
\| \boldsymbol{w}(k\eta) - \boldsymbol{w}_k \| \leq O(\eta). \tag{21}
$$

*Proof.* Let

$$
\bar{\boldsymbol{w}}(k\eta) = \boldsymbol{w}((k-1)\eta) - \eta \nabla \mathcal{L}(\boldsymbol{w}((k-1)\eta)). \tag{22}
$$

Then by Lipschitz gradient of $\mathcal{L}(\cdot)$, we see

$$
\begin{aligned}
\|\boldsymbol{w}(k\eta) - \bar{\boldsymbol{w}}(k\eta)\| &= \left\| \int_{(k-1)\eta}^{k\eta} \nabla\mathcal{L}(\boldsymbol{w}(u)) - \nabla\mathcal{L}(\boldsymbol{w}((k-1)\eta))du \right\| \\
&\leq \int_{(k-1)\eta}^{k\eta} L \|\boldsymbol{w}(u) - \boldsymbol{w}((k-1)\eta)\| \, du \\
&= \int_{(k-1)\eta}^{k\eta} L \left\| \int_{(k-1)\eta}^{u} \nabla\mathcal{L}(\boldsymbol{w}(s))ds \right\| du \\
&= O(\eta^2).
\end{aligned}
\tag{23}
$$

Then we have

$$
\begin{aligned}
\|\boldsymbol{w}(k\eta) - \boldsymbol{w}_k\| &\leq \|\boldsymbol{w}_k - \bar{\boldsymbol{w}}(k\eta)\| + \|\boldsymbol{w}(k\eta) - \bar{\boldsymbol{w}}(k\eta)\| \\
&\leq \|\boldsymbol{w}((k-1)\eta) - \boldsymbol{w}_{k-1}\| + \eta\|\nabla\mathcal{L}(\boldsymbol{w}((k-1)\eta)) - \nabla\mathcal{L}(\boldsymbol{w}_{k-1})\| + O(\eta^2) \\
&\leq (1 + \eta L)\|\boldsymbol{w}((k-1)\eta) - \boldsymbol{w}_{k-1}\| + O(\eta^2) \\
&\leq \cdots \leq (1 + \eta L)^k \|\boldsymbol{w}(0) - \boldsymbol{w}_0\| + O(\eta) \\
&= O(\eta),
\end{aligned}
\tag{24}
$$

for $\boldsymbol{w}(0) = \boldsymbol{w}_0$, which results in the conclusion. $\qquad\square$

Before proving the Theorem 1, we give a lemma to illustrate that $\boldsymbol{w}(t)$ will converge to the direction of $\hat{\boldsymbol{w}}$.

**Lemma 3.** *There exists a $t_0$, for $t > t_0$, we have*

$$
\boldsymbol{w}(t) = \rho(t)\hat{\boldsymbol{w}} + \boldsymbol{h}(t),
\tag{25}
$$

*for some $\rho(t)$ and $\boldsymbol{h}(t)$. Here $\rho(t)$ goes to infinity and $\|\boldsymbol{h}(t)\|$ is in the order of $o(\rho(t))$.*

*Proof.* Let

$$
\boldsymbol{r}(t) = \boldsymbol{w}(t) - \frac{\|\boldsymbol{w}(t)\|}{\|\hat{\boldsymbol{w}}\|}\hat{\boldsymbol{w}},
\tag{26}
$$

$\|\boldsymbol{r}(t)\| = o(\|\boldsymbol{w}(t)\|)$ will implies the conclusion. Since $\ell'(\cdot)$ has exponential tail Assumption 2, there exists a $t_0$, for $t > t_0$, we have

$$
\begin{aligned}
\frac{d}{dt}\|\boldsymbol{r}(t)\| &= \frac{\dot{\boldsymbol{r}}^T(t)\boldsymbol{r}(t)}{\|\boldsymbol{r}(t)\|} = -\frac{1}{\|\boldsymbol{r}(t)\|}\left(1 - \frac{\boldsymbol{w}(t)^T\hat{\boldsymbol{w}}}{\|\boldsymbol{w}(t)\|\|\hat{\boldsymbol{w}}\|}\right)\nabla\mathcal{L}(\boldsymbol{w}(t))^T\boldsymbol{r}(t) \\
&\leq \frac{C}{N\|\boldsymbol{r}(t)\|}(1 - \cos(\boldsymbol{w}(t), \hat{\boldsymbol{w}}))\sum_{i=1}^{N}\exp\left(-\boldsymbol{w}(t)^T\boldsymbol{x}_i + \varepsilon\|\boldsymbol{w}(t)\|\right)\left(\boldsymbol{x}_i - \varepsilon\frac{\boldsymbol{w}(t)}{\|\boldsymbol{w}(t)\|}\right)^T\boldsymbol{r}(t).
\end{aligned}
\tag{27}
$$

Then, we see

$$
\begin{aligned}
&\sum_{i=1}^{N}\exp\left(-\boldsymbol{w}(t)^T\boldsymbol{x}_i + \varepsilon\|\boldsymbol{w}(t)\|\right)\left(\boldsymbol{x}_i - \varepsilon\frac{\boldsymbol{w}(t)}{\|\boldsymbol{w}(t)\|}\right)^T\boldsymbol{r}(t) \\
&= \sum_{i=1}^{N}\exp\left(-\boldsymbol{r}(t)^T\boldsymbol{x}_i - \frac{\|\boldsymbol{w}(t)\|\hat{\boldsymbol{w}}^T\boldsymbol{x}_i}{\|\hat{\boldsymbol{w}}\|} + \varepsilon\|\boldsymbol{w}(t)\|\right)\left(\boldsymbol{x}_i - \varepsilon\frac{\boldsymbol{w}(t)}{\|\boldsymbol{w}(t)\|}\right)^T\boldsymbol{r}(t) \\
&\leq \sum_{i=1}^{N}\exp\left(-\frac{\|\boldsymbol{w}(t)\|\hat{\boldsymbol{w}}^T\boldsymbol{x}_i}{\|\hat{\boldsymbol{w}}\|} + \varepsilon\|\boldsymbol{w}(t)\|\right) - \varepsilon\exp\left(-\boldsymbol{r}(t)^T\boldsymbol{x}_i - \frac{\|\boldsymbol{w}(t)\|\hat{\boldsymbol{w}}^T\boldsymbol{x}_i}{\|\hat{\boldsymbol{w}}\|} + \varepsilon\|\boldsymbol{w}(t)\|\right)\frac{\boldsymbol{w}(t)^T}{\|\boldsymbol{w}(t)\|}\boldsymbol{r}(t) \\
&\leq N\exp\left(-(\varepsilon_1 - \varepsilon)\|\boldsymbol{w}(t)\|\right),
\end{aligned}
\tag{28}
$$

where $\hat{\boldsymbol{w}}$ is the hard margin classifier and $\varepsilon$ is the distance between support vectors and $\hat{\boldsymbol{w}}$. Here we use the relationship that $ze^{-z} \leq 1$ for any $z$ and $\boldsymbol{w}(t)^T\boldsymbol{r}(t) \geq 0$. Since

$$
(1 - \cos(\boldsymbol{w}(t), \hat{\boldsymbol{w}})) = -\frac{\boldsymbol{r}(t)^T\hat{\boldsymbol{w}}}{\|\boldsymbol{w}(t)\|\|\hat{\boldsymbol{w}}\|},
\tag{29}
$$

and $\boldsymbol{r}(t)^T \hat{\boldsymbol{w}} \leq 0$, we have

$$\frac{d}{dt}\|\boldsymbol{r}(t)\| \leq -CN \frac{\boldsymbol{r}(t)^T \hat{\boldsymbol{w}}}{\|\boldsymbol{r}(t)\|\|\boldsymbol{w}(t)\|\|\hat{\boldsymbol{w}}\|} \exp\left(-(\varepsilon_1 - \varepsilon)\|\boldsymbol{w}(t)\|\right) \leq \frac{CN}{\|\boldsymbol{w}(t)\|} \exp\left(-(\varepsilon_1 - \varepsilon)\|\boldsymbol{w}(t)\|\right). \quad (30)$$

Similar to Theorem 3 in Su et al. (2015), by exponential tail of $l(\cdot)$ Assumption 2, we have

$$C_1 \exp\left(-\boldsymbol{w}(t)^T \boldsymbol{x}_i + \varepsilon\|\boldsymbol{w}(t)\|\right) \leq \ell\left(\boldsymbol{w}(t)^T \boldsymbol{x}_i - \varepsilon\|\boldsymbol{w}(t)\|\right) \leq \frac{N\|\hat{\boldsymbol{w}} - \boldsymbol{w}_0\|}{2t}, \quad (31)$$

for some constant $C_1$ and any $\boldsymbol{x}_i$. Then we see

$$\log \frac{2C_1}{N\|\hat{\boldsymbol{w}} - \boldsymbol{w}_0\|} + \log t \leq \boldsymbol{w}(t)^T \boldsymbol{x}_i - \varepsilon\|\boldsymbol{w}(t)\|. \quad (32)$$

On the other hand, by the definition of hard margin classifier and Lemma 1, there exists $\boldsymbol{x}_i$ such that $\boldsymbol{w}(t)^T \boldsymbol{x}_i \leq \varepsilon_1\|\boldsymbol{w}(t)\|$. Combining this and equation 32, we have

$$\frac{1}{\varepsilon_1 - \varepsilon}\left(\log \frac{C_1}{N\|\hat{\boldsymbol{w}} - \boldsymbol{w}_0\|} + \log t\right) \leq \|\boldsymbol{w}(t)\|. \quad (33)$$

Plugging this into equation 30, and we see that there exists a $t_1$ such that $\frac{1}{2}\log t \geq \log \frac{N\|\hat{\boldsymbol{w}}-\boldsymbol{w}_0\|}{C_1}$, then

$$\|\boldsymbol{r}(t)\| \leq \|\boldsymbol{r}(t_0 \wedge t_1)\| + \frac{(\varepsilon_1 - \varepsilon)N^2\|\hat{\boldsymbol{w}} - \boldsymbol{w}_0\|}{CC_1} \int_{t_0 \wedge t_1}^{t} \frac{1}{t \log t} dt = O(\log\log t), \quad (34)$$

for $t \geq t_0 \wedge t_1$. Combining these, we conclude that $\|\boldsymbol{r}(t)\|$ is in the order of $o(\|\boldsymbol{w}(t)\|)$. $\qquad\square$

Next, we use a lemma to illustrate the explicit order of $\rho(t)$

**Lemma 4.** $\rho(t)$ in Lemma 3 is on the scale of $\frac{\log t}{\varepsilon_1 - \varepsilon}$, where $\{\varepsilon_i\}$ is the sorted distance of $\{\boldsymbol{x}_i\}_{i=1}^N$ to hard margin solution of SVM $\hat{\boldsymbol{w}}$.

*Proof.* From Lemma 3 and 3, we have

$$\boldsymbol{w}(t) = \rho(t)\frac{\hat{\boldsymbol{w}}}{\|\hat{\boldsymbol{w}}\|} + \boldsymbol{h}(t), \quad (35)$$

where $\rho(t) \to \infty$ and $\|\boldsymbol{h}(t)\| = o(\rho(t))$. Specifically, the $\boldsymbol{h}(t)$ can be chosen to be orthogonal with $\hat{\boldsymbol{w}}$, otherwise we can use a decomposition in a direct sum. For $\|\boldsymbol{w}(t)\|$, we have

$$\begin{aligned}
\|\boldsymbol{w}(t)\| &= \sqrt{\rho(t)^2 + \|\boldsymbol{h}(t)\|^2} \\
&= \sqrt{\rho(t)^2 + \|\boldsymbol{h}(t)\|^2} - \rho(t) + \rho(t) \\
&= \frac{\left(\sqrt{\rho(t)^2 + \|\boldsymbol{h}(t)\|^2} + \rho(t)\right)\left(\sqrt{\rho(t)^2 + \|\boldsymbol{h}(t)\|^2} - \rho(t)\right)}{\sqrt{\rho(t)^2 + \|\boldsymbol{h}(t)\|^2} + \rho(t)} + \rho(t) \\
&= \frac{\|\boldsymbol{h}(t)\|^2}{\rho(t) + \sqrt{\rho(t)^2 + \|\boldsymbol{h}(t)\|^2}} + \rho(t).
\end{aligned} \quad (36)$$

On the other hand, for $t$ large enough,

$$\begin{aligned}
\frac{d}{dt}\|\boldsymbol{w}(t)\| &= -\nabla\mathcal{L}(\boldsymbol{w}(t))^T \frac{\boldsymbol{w}(t)}{\|\boldsymbol{w}(t)\|} \\
&\leq \frac{C_1}{N}\sum_{i=1}^N \exp\left(-\boldsymbol{w}(t)^T \boldsymbol{x}_i + \varepsilon\|\boldsymbol{w}(t)\|\right)\left(\boldsymbol{x}_i - \varepsilon\frac{\boldsymbol{w}(t)}{\|\boldsymbol{w}(t)\|}\right)^T \frac{\boldsymbol{w}(t)}{\|\boldsymbol{w}(t)\|} \\
&= \frac{C_1}{N}\sum_{i=1}^N \exp\left(-\rho(t)\frac{\hat{\boldsymbol{w}}^T \boldsymbol{x}_i}{\|\hat{\boldsymbol{w}}\|} + \boldsymbol{h}(t)^T \boldsymbol{x}_i + \varepsilon\rho(t) + \varepsilon\frac{\|\boldsymbol{h}(t)\|^2}{\rho(t) + \sqrt{\rho(t)^2 + \|\boldsymbol{h}(t)\|^2}}\right)\left(\frac{\hat{\boldsymbol{w}}^T \boldsymbol{x}_i - \varepsilon\|\hat{\boldsymbol{w}}\|}{\|\hat{\boldsymbol{w}}\|}\right) \\
&\leq \frac{C_2}{N}\sum_{i=1}^N \exp\left(-(\varepsilon_i - \varepsilon)\rho(t)\right)(\varepsilon_i - \varepsilon).
\end{aligned}$$

$$(37)$$

for some constant $C_1, C_2$. Since $\rho(t)$ goes to infinity and $\|\boldsymbol{h}(t)\| = o(\rho(t))$, the derivation of $\frac{\|\boldsymbol{h}(t)\|^2}{\rho(t)+\sqrt{\rho(t)^2+\|\boldsymbol{h}(t)\|^2}}$ will $o(\rho'(t))$. Similar to equation 37, we can get the lower bound of $\frac{d}{dt}\|\boldsymbol{w}(t)\|$. In summary, we have

$$\frac{C_3}{N} \exp\left(-(\varepsilon_1 - \varepsilon)\rho(t)\right)(\varepsilon_1 - \varepsilon) \leq \rho'(t) \leq C_2 \exp\left(-(\varepsilon_1 - \varepsilon)\rho(t)\right)(\varepsilon_N - \varepsilon) \tag{38}$$

for constant some $C_3$. Hence we can conclude that

$$\frac{\log t}{\varepsilon_1 - \varepsilon} + C_5 \leq \rho(t) \leq \frac{\log t}{\varepsilon_1 - \varepsilon} + C_4, \tag{39}$$

for some constant $C_4, C_5$. It results in $\rho(t) = O(\log t)$. $\qquad\square$

We have proven that $\boldsymbol{w}(t)$ will converge to the direction of $\hat{\boldsymbol{w}}$. Now, we will show $\boldsymbol{w}(t)$ will have the same support vector with $\hat{\boldsymbol{w}}$ when $t$ is large. It's a key fact of proving Theorem 1.

**Lemma 5.** *There exists $t_0$ such that $\boldsymbol{w}(t)$ will have the same support vectors with $\hat{\boldsymbol{w}}$ for $t > t_0$.*

*Proof.* Let $\mathbf{S} = \{i : \hat{\boldsymbol{w}}^T \boldsymbol{x}_i = \varepsilon_1 \|\hat{\boldsymbol{w}}\|\}$, for $i \in \mathbf{S}, j \notin \mathbf{S}$, we have

$$\begin{aligned}
\frac{\boldsymbol{w}(t)^T}{\|\boldsymbol{w}(t)\|}(\boldsymbol{x}_i - \boldsymbol{x}_j) &= \frac{\left(\rho(t)\frac{\hat{\boldsymbol{w}}}{\|\hat{\boldsymbol{w}}\|} + \boldsymbol{h}(t)\right)^T}{\|\boldsymbol{w}(t)\|}(\boldsymbol{x}_i - \boldsymbol{x}_j) \\
&\leq \frac{1}{\|\boldsymbol{w}(t)\|}\left(\rho(t)\frac{\hat{\boldsymbol{w}}^T}{\|\hat{\boldsymbol{w}}\|}(\boldsymbol{x}_i - \boldsymbol{x}_j) + \|\boldsymbol{h}(t)\|\|\boldsymbol{x}_i - \boldsymbol{x}_j\|\right)
\end{aligned} \tag{40}$$

Since $\boldsymbol{x}_i$ is the support vector of $\hat{\boldsymbol{w}}$, $\frac{\hat{\boldsymbol{w}}^T}{\|\hat{\boldsymbol{w}}\|}(\boldsymbol{x}_i - \boldsymbol{x}_j) > 0$. Then, $\|\boldsymbol{h}(t)\| = o(\rho(t))$ shows that $\frac{\boldsymbol{w}(t)^T}{\|\boldsymbol{w}(t)\|}(\boldsymbol{x}_i - \boldsymbol{x}_j) \leq 0$ for a large $t$. Then we get the conclusion. $\qquad\square$

## C    PROOF OF THEOREM 1

In this section, we give a fully characterization to the proof of Theorem 1.

**Restate of Theorem 1.** *For any $\varepsilon$-strongly linearly separable data (Definition 1), and loss function $\ell(\cdot)$ satisfies Assumption 1 and 2. If the Assumption 3 holds, then the gradient flow iterates $\boldsymbol{w}(t)$,*

$$\frac{d\boldsymbol{w}(t)}{dt} = -\nabla\mathcal{L}(\boldsymbol{w}(t)) \tag{41}$$

*satisfies*

$$\boldsymbol{w}(t) = \hat{\boldsymbol{w}} \cdot O(\log t) + \boldsymbol{h}(t), \tag{42}$$

*for a large $t$. Here $\hat{\boldsymbol{w}}$ is the hard margin solution of SVM:*

$$\hat{\boldsymbol{w}} = \arg\min_{\boldsymbol{w}\in\mathbb{R}^d} \|\boldsymbol{w}\| \qquad \text{s.t. } \boldsymbol{w}^T\boldsymbol{x}_i \geq 1. \tag{43}$$

$\|\boldsymbol{h}(t)\|$ *is in the order of $o(\log t)$. Then*

$$\lim_{t\to\infty}\left\|\frac{\boldsymbol{w}(t)}{\|\boldsymbol{w}(t)\|} - \frac{\hat{\boldsymbol{w}}}{\|\hat{\boldsymbol{w}}\|}\right\| \leq O((\log t)^{-(1+\varepsilon^*)}), \tag{44}$$

*where $\varepsilon^* = \frac{|\mathbf{S}|\varepsilon(\varepsilon_1 - \varepsilon)}{N}$, $\{\varepsilon_i\}$ is the sorted distance of data away from the hard margin solution of SVM, and $\mathbf{S} = \{i : \hat{\boldsymbol{w}}^T\boldsymbol{x}_i = \varepsilon_1\|\hat{\boldsymbol{w}}\|\}$; $|\mathbf{S}|$ is the number of elements in set $\mathbf{S}$. Finally, iterates $\{\boldsymbol{w}_t\}$ of adversarial training updated by GD with step size $\eta$ will satisfy*

$$\|\boldsymbol{w}_k - \boldsymbol{w}(k\eta)\| \leq O(\eta), \tag{45}$$

*for $k \in \mathbb{N}^+$. Then we can conclude*

$$\lim_{t\to\infty}\left\|\frac{\boldsymbol{w}_t}{\|\boldsymbol{w}_t\|} - \frac{\hat{\boldsymbol{w}}}{\|\hat{\boldsymbol{w}}\|}\right\| \leq \lim_{t\to\infty}\left\|\frac{\boldsymbol{w}(t\eta)}{\|\boldsymbol{w}(t\eta)\|} - \frac{\hat{\boldsymbol{w}}}{\|\hat{\boldsymbol{w}}\|}\right\| + O(\eta). \tag{46}$$

*Proof.* Let $\boldsymbol{w}(t) = \rho(t)\frac{\hat{\boldsymbol{w}}}{\|\hat{\boldsymbol{w}}\|} + \boldsymbol{h}(t)$. Denoting the support vectors set by $\mathbf{S}$, $\mathbf{S} = \{i : \hat{\boldsymbol{w}}^T\boldsymbol{x}_i = \varepsilon_1\|\hat{\boldsymbol{w}}\|\}$. By the exponential tail of $\ell'(u)$ (Assumption 2), there exists a $t_0$, for $t > t_0$, we have

$$
\frac{1}{2}\frac{d}{dt}\|\boldsymbol{h}(t)\|^2
$$

$$
= -\left(\nabla\mathcal{L}(\boldsymbol{w}(t)) + \rho'(t)\frac{\hat{\boldsymbol{w}}}{\|\hat{\boldsymbol{w}}\|}\right)^T \boldsymbol{h}(t) = -\nabla\mathcal{L}(\boldsymbol{w}(t))^T\boldsymbol{h}(t)
$$

$$
\leq \frac{C}{N}\left(\sum_{i\in\mathbf{S}} + \sum_{i\notin\mathbf{S}}\right)\exp\left(-\boldsymbol{w}(t)\boldsymbol{x}_i + \varepsilon\|\boldsymbol{w}(t)\|\right)\left(\boldsymbol{x}_i - \varepsilon\frac{\boldsymbol{w}(t)}{\|\boldsymbol{w}(t)\|}\right)^T \boldsymbol{h}(t)
$$

$$
= \frac{C}{N}\left(\sum_{i\in\mathbf{S}} + \sum_{i\notin\mathbf{S}}\right)\exp\left(-\rho(t)\frac{\hat{\boldsymbol{w}}^T\boldsymbol{x}_i}{\|\hat{\boldsymbol{w}}\|} - \boldsymbol{h}(t)^T\boldsymbol{x}_i + \varepsilon\rho(t) + \frac{\|\boldsymbol{h}(t)\|^2}{\rho(t) + \sqrt{\rho(t)^2 + \|\boldsymbol{h}(t)\|^2}}\right)\left(\boldsymbol{x}_i - \varepsilon\frac{\boldsymbol{w}(t)}{\|\boldsymbol{w}(t)\|}\right)^T \boldsymbol{h}(t)
$$

$$
\leq \frac{C}{N}\sum_{i\in\mathbf{S}}\exp\left(-\rho(t)\frac{\hat{\boldsymbol{w}}^T\boldsymbol{x}_i}{\|\hat{\boldsymbol{w}}\|} - \boldsymbol{h}(t)^T\boldsymbol{x}_i + \varepsilon\rho(t)\right)\left(\boldsymbol{x}_i - \varepsilon\frac{\boldsymbol{w}(t)}{\|\boldsymbol{w}(t)\|}\right)^T \boldsymbol{h}(t)
$$

$$
\leq -\frac{\varepsilon}{N}\sum_{i\in\mathbf{S}}\exp\left(-\|\boldsymbol{h}(t)\|\|\boldsymbol{x}_i\| - (\varepsilon_i - \varepsilon)\rho(t)\right)\frac{\|\boldsymbol{h}(t)\|^2}{\rho(t)},
$$

(47)

[7]Here we use a fact that $\boldsymbol{x}_i^T\boldsymbol{h}(t) \leq 0$ for $i \in \mathbf{S}$, due to $\hat{\boldsymbol{w}}$ is the hard margin classifier. In addition,

$$
1 + \frac{\|\boldsymbol{h}(t)\|^2}{\rho(t) + \sqrt{\rho(t)^2 + \|\boldsymbol{h}(t)\|^2}} \leq \exp\left(\frac{\|\boldsymbol{h}(t)\|^2}{\rho(t) + \sqrt{\rho(t)^2 + \|\boldsymbol{h}(t)\|^2}}\right). \tag{48}
$$

With this, we can choose a $t_0$ such that

$$
\sum_{i\in\mathbf{S}}\exp\left(-\rho(t)\frac{\hat{\boldsymbol{w}}^T\boldsymbol{x}_i}{\|\hat{\boldsymbol{w}}\|} - \boldsymbol{h}(t)^T\boldsymbol{x}_i + \varepsilon\rho(t)\right)\left(\boldsymbol{x}_i - \varepsilon\frac{\boldsymbol{w}(t)}{\|\boldsymbol{w}(t)\|}\right)^T \boldsymbol{h}(t)\frac{\|\boldsymbol{h}(t)\|^2}{\rho(t) + \sqrt{\rho(t)^2 + \|\boldsymbol{h}(t)\|^2}}
$$

$$
+ \sum_{i\notin\mathbf{S}}\exp\left(-\boldsymbol{w}(t)\boldsymbol{x}_i + \varepsilon\|\boldsymbol{w}(t)\|\right)\left(\boldsymbol{x}_i - \varepsilon\frac{\boldsymbol{w}(t)}{\|\boldsymbol{w}(t)\|}\right)^T \boldsymbol{h}(t) \leq 0,
$$

(49)

for $t > t_0$. This concludes the equation equation 47. Let $\|\boldsymbol{h}(t)\|\|\boldsymbol{x}_i\|$ be $C(t)$, we can first derive that $\|\mathrm{h}(t)\|$ will converge to zero, then $C(t)$ can be close to zero. By Gronwall's inequality, we have

$$
\|\boldsymbol{h}(t)\|^2 \leq \|\boldsymbol{h}(t_0)\|^2 \exp\left(\int_{t_0}^t -\frac{\varepsilon e^{-C(u)}}{N\rho(u)}\sum_{i\in\mathbf{S}}\exp\left(-(\varepsilon_i - \varepsilon)\rho(u)\right)du\right), \tag{50}
$$

$C(t)$ can be arbitrary small when $t > t_0$. Plugging this into equation 50, and combining Lemma 4, we have

$$
\|\boldsymbol{h}(t)\|^2 \leq \|\boldsymbol{h}(t_0)\|^2 \exp\left(\int_{t_0}^t -\frac{\varepsilon(\varepsilon_1 - \varepsilon)}{N\log u}\sum_{i\in\mathbf{S}}u^{-\left(\frac{\varepsilon_i-\varepsilon}{\varepsilon_1-\varepsilon}\right)}du\right)
$$

$$
= \|\boldsymbol{h}(t_0)\|^2 \exp\left(-\sum_{i\in\mathbf{S}}\frac{\varepsilon(\varepsilon_1 - \varepsilon)}{N}\int_{t_0}^t \frac{1}{u\log u}du\right) \tag{51}
$$

$$
= \|\boldsymbol{h}(t_0)\|^2 \exp\left(-\frac{|\mathbf{S}|\varepsilon(\varepsilon_1 - \varepsilon)}{N}\log\log t\right)
$$

$$
= O(\log t)^{-\varepsilon^*},
$$

where $\varepsilon^* = \frac{|\mathbf{S}|\varepsilon(\varepsilon_1-\varepsilon)}{N}$. Since we have

$$
\frac{\boldsymbol{w}(t)}{\|\boldsymbol{w}(t)\|} = \frac{\rho(t)}{\rho(t) + \frac{\|\boldsymbol{h}(t)\|^2}{\rho(t)}}\frac{\hat{\boldsymbol{w}}}{\|\hat{\boldsymbol{w}}\|} + \frac{\boldsymbol{h}(t)}{\rho(t) + \frac{\|\boldsymbol{h}(t)\|^2}{\rho(t)}}, \tag{52}
$$

---

[7]Here we hide the constant $C$ in the last inequality, which is decided by loss function $l(\cdot)$. $C$ will usually smaller than 1 in fact.

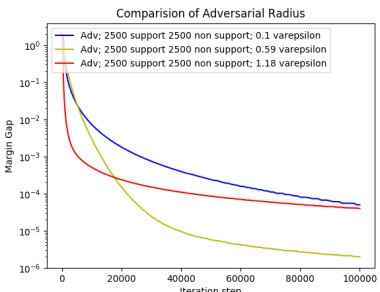 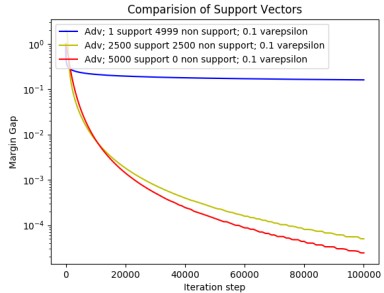

(a) Adversarial training with different $\varepsilon$.  (b) Adversarial training with different number of support vectors.

Figure 5: Gap between the direction of iterates and direction of hard margin classifier in $l_4$ space.

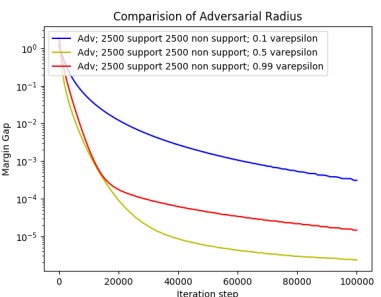 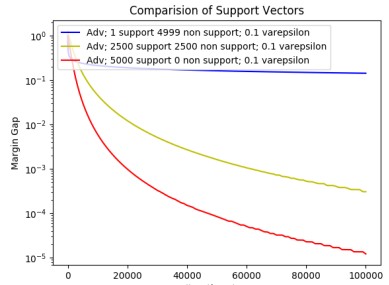

(a) Adversarial training with different $\varepsilon$.  (b) Adversarial training with different number of support vectors.

Figure 6: Gap between the direction of iterates and direction of hard margin classifier in $l_\infty$ space.

then we see

$$
\begin{aligned}
\left\| \frac{\boldsymbol{w}(t)}{\|\boldsymbol{w}(t)\|} - \frac{\hat{\boldsymbol{w}}}{\|\hat{\boldsymbol{w}}\|} \right\| &\leq \left(1 - \frac{\rho(t)}{\rho(t) + \frac{\|\boldsymbol{h}(t)\|^2}{\rho(t)}}\right) + \frac{\|\boldsymbol{h}(t)\|}{\rho(t) + \frac{\|\boldsymbol{h}(t)\|^2}{\rho(t)}} \\
&= \frac{\|\boldsymbol{h}(t)\|^2}{\rho(t)^2 + \|\boldsymbol{h}(t)\|^2} + \frac{\|\boldsymbol{h}(t)\|}{\rho(t) + \frac{\|\boldsymbol{h}(t)\|^2}{\rho(t)}} \\
&= O\left(\log\right)^{-(1+\varepsilon^*)}
\end{aligned}
\tag{53}
$$

when $t$ goes to infinity. Since $\boldsymbol{h}(t) \leq O(\log t)^{-\varepsilon^*}$, for $\varepsilon^* = \frac{|\mathbf{S}|\varepsilon(\varepsilon_1-\varepsilon)}{N}$. Combining Lemma 2, we can get the conclusion. $\qquad\square$

## D  PROOF OF THEOREM 2

*Proof.* The $\boldsymbol{x}_{k,i}$ will either satisfies $|\boldsymbol{w}^{*T}\boldsymbol{x}_{k,i}| \leq \varepsilon\|\boldsymbol{w}^*\|$ nor $|\boldsymbol{w}^{*T}\boldsymbol{x}_{k,i}| \geq \varepsilon\|\boldsymbol{w}^*\|$. We know the first inequality represents the distance between $\boldsymbol{x}_{k,i}$ and $\boldsymbol{w}^*$ is smaller than $\varepsilon$. Then for $\boldsymbol{w}^{*T}\boldsymbol{x}_{k,i} \geq \varepsilon\|\boldsymbol{w}^*\|$, due to $\varepsilon \geq \frac{1}{2}\max_{i,j}\|\boldsymbol{x}_{k,i} - \boldsymbol{x}_{k',j}\|$ for any $i,j$, there exists $\boldsymbol{x}'_{k',j} \in B_2(\boldsymbol{x}_{k',j},\varepsilon)\bigcap B_2(\boldsymbol{x}_{k,i},\varepsilon)$ with $k',k = 1,2$ and $k' \neq k$. By triangle inequality, the distance between $\boldsymbol{x}'_{k',j}$ and $\boldsymbol{w}^*$ is smaller than $\frac{\boldsymbol{w}^{*T}\boldsymbol{x}_{k,i}}{\|\boldsymbol{w}^*\|} - \varepsilon$. Since $\boldsymbol{x}_{k',j}$ locates in different category with $\boldsymbol{x}_{k,i}$, by the monotone decreasing property of $\ell(\cdot)$, we have

$$
\ell(-\boldsymbol{w}^{*T}\boldsymbol{x}_{k,i} + \varepsilon\|\boldsymbol{w}^*\|) \leq \ell(\boldsymbol{x}'_{k'j}) \leq \max_{\|\boldsymbol{x}-\boldsymbol{x}_{k',j}\|\leq\varepsilon} \ell(\boldsymbol{w}^{*T}\boldsymbol{x}).
\tag{54}
$$

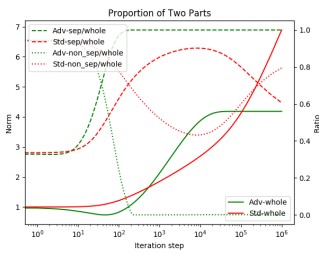 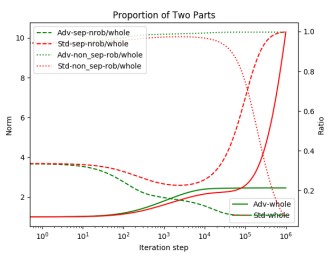 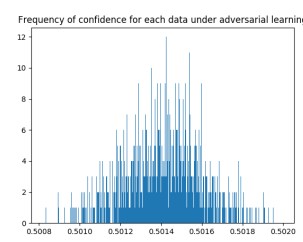

(a) Non-$\varepsilon$-strongly linearly separable but linearly separable data.

(b) Non-linearly separable data.

(c) Frequency of confidence for data point under adversarial training

Figure 7: Adversarial training in $l_4$ space. Outliers are added into dataset of the first two subfigure. The third subfigure is confidence distribution for adversarial training when data are linearly separable but $\varepsilon \geq \frac{1}{2} \max_{i,j} \|\boldsymbol{x}_i - \boldsymbol{x}_j\|$. The $y$-axis in the third pictures is number of data.

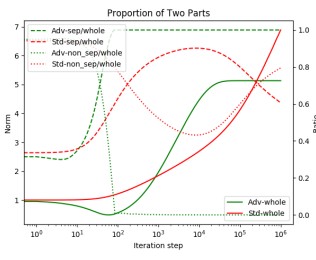 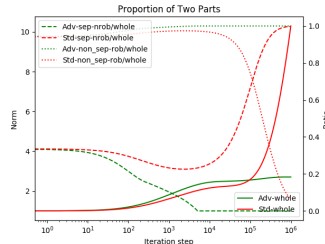 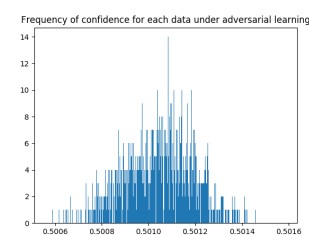

(a) Non-$\varepsilon$-strongly linearly separable but linearly separable data.

(b) Non-linearly separable data.

(c) Frequency of confidence for data point under adversarial training

Figure 8: Adversarial training in $l_\infty$ space. Outliers are added into dataset of the first two subfigure. The third subfigure is confidence distribution for adversarial training when data are linearly separable but $\varepsilon \geq \frac{1}{2} \max_{i,j} \|\boldsymbol{x}_i - \boldsymbol{x}_j\|$. The $y$-axis in the third pictures is number of data.

Hence we have

$$\ell(-\boldsymbol{w}^{*T}\boldsymbol{x}_{k,i} + \varepsilon\|\boldsymbol{w}^*\|) \leq \frac{1}{N_{k'}} \sum_{j=1}^{N_{k'}} \max_{\|\boldsymbol{x}-\boldsymbol{x}_{k',j}\|\leq\varepsilon} \ell(\boldsymbol{w}^{*T}\boldsymbol{x} - \varepsilon\|\boldsymbol{w}^*\|). \tag{55}$$

Then, we have

$$N_2\ell\left(-|\boldsymbol{w}^{*T}\boldsymbol{x}_{1,i}| + \varepsilon\|\boldsymbol{w}^*\|\right) + N_1\ell\left(-|\boldsymbol{w}^{*T}\boldsymbol{x}_{2,i}| + \varepsilon\|\boldsymbol{w}^*\|\right) \leq (N_1 + N_2)\ell(0), \tag{56}$$

if $\boldsymbol{w}^*$ can correctly predict a point far from margin larger than $\varepsilon$ in each category, for those $\boldsymbol{w}^{*T}\boldsymbol{x}_{k,i} \geq \varepsilon\|\boldsymbol{w}^*\|$ due to $\boldsymbol{w}^*$ is minimum, then $\mathcal{L}(\boldsymbol{w}^*) \leq \mathcal{L}(\boldsymbol{0})$. On the other hand, for $\boldsymbol{w}^{*T}\boldsymbol{x}_{k,i} \leq -\varepsilon\|\boldsymbol{w}^*\|$, we can immediately derive

$$\ell(\boldsymbol{w}^{*T}\boldsymbol{x}_{k,i}) \leq \ell(-\boldsymbol{w}^{*T}\boldsymbol{x}_{k',j} - \varepsilon\|\boldsymbol{w}^*\|), \tag{57}$$

by triangle inequality where $\boldsymbol{w}^{*T}\boldsymbol{x}_{k',j} \geq 0$. Then we get the conclusion. □

# E EXTRA EXPERIMENTS

## E.1 ADVERSARIAL TRAINING IN $l_p$ SPACE

Our conclusions are obtained for adversarial training in $l_2$ space while it can be conducted in a more general $l_p$ space (Goodfellow et al., 2014; Madry et al., 2018). It is meaningful to verify whether these conclusions are still hold in $l_p$ space. Hence, we extend our experiments to $l_p$ space.

The adversarial training objective within linear classifier can be formulated as

$$\mathcal{L}(\boldsymbol{w}) = \frac{1}{N} \sum_{i=1}^{N} \max_{\|\boldsymbol{x}-\boldsymbol{x}_i\|_p \leq \varepsilon} \ell(\boldsymbol{w}^T \boldsymbol{x}), \tag{58}$$

in $l_p$ space, where $\|\cdot\|_p$ is $l_p$ norm. It has an explicit formulation

$$\mathcal{L}(\boldsymbol{w}) = \frac{1}{N} \sum_{i=1}^{N} \ell\left(\boldsymbol{w}^T \boldsymbol{x}_i - \varepsilon \|\boldsymbol{w}\|_q\right), \tag{59}$$

where $\frac{1}{p} + \frac{1}{q} = 1$. We conduct experiments for $p = 4$ and $p = \infty$.

We empirically verify that adversarial training in $l_p$ space helps accelerating the convergence to hard margin solution of SVM on $\varepsilon$-strongly linearly separable data. Besides that, adversarial training can benefit from more support vectors and an appropriate adversarial radius $\varepsilon$. For non-$\varepsilon$-strongly linearly separable data, we try to validate that adversarial training in $l_p$ space is stable to outliers while it will obtain a classifier with low confidence on each data. All the experiments we conducted here are following the settings in Section 5, expect for the $p$ are respectively chosen as 4 and $\infty$. In addition, we should notice that the distance between support vectors locate in $\mathbf{1}_d^T \boldsymbol{x} = \pm d$ and hard margin classifier $\mathbf{1}_d$ in $l_p$ space is $\frac{d}{\|\mathbf{1}_d\|_q}$. Hence we should adjust the adversarial radius $\varepsilon$ for different $p$.

We first verify the conclusion for $\varepsilon$-strongly linearly separable data. The results for $p = 4$ and $p = \infty$ can be respectively referred to Figure 5 and 6. Noticing that the distance between support vectors and hard margin solution of SVM is $\frac{d}{\|\mathbf{1}_d\|_q} = d^{\frac{1}{p}}$. Then for $p = 4$, $p = \infty$ and $d = 2$, we have $2^{\frac{1}{4}} \approx 1.18$, $2^0 = 1$. From the results, we see that an appropriate adversarial radius ($\varepsilon = \varepsilon_1/2$) and more support vectors are helpful for adversarial training even in $l_p$ space.

Now, we pay our attention to non-$\varepsilon$-strongly linearly separable data. We respectively verify our conclusions about stability and confidence of adversarial training. The experimental settings are again same with Section 5. The results can be referred to Figure 7 and 8. We see that our conclusions are still hold in $l_p$ space, i.e. classifier obtained by adversarial training is stable to outliers and the confidence of it will consist keep in a low level for a large $\varepsilon$.

As a matter of fact, we can generalize our theoretical results to $l_p$ space by following the methods in this paper. The inner product $\langle \cdot, \cdot \rangle$ of linear classifier is derived from $\|\cdot\|_2$ while $l_2$ is a Hilbert space, but $l_p$ space is not. The inner product we used does not math the $l_p$ space. Hence, we need adding some extra bounded conditions to extend our conclusions to $l_p$ space.

### E.2 ADVERSARIAL TRAINING FOR NEURAL NETWORK

Our conclusions are derived from linear classifier, but Du et al. (2019); Jacot et al. (2018); Lee et al. (2019b); Arora et al. (2019) suggest that over-parameterized neural networks of sufficient width (or infinite width) evolve as linear models with Neural Tangent Kernel (NTK). Hence our conclusions can somehow represent the performance of adversarial training for neural network models. In this subsection, we try to empirically verify our conclusions about adversarial training within neural network model.

We use CIFAR10 (Krizhevsky et al., 2012) to conduct our experiments. CIFAR10 is a dataset with 10 categories. To simplify the experiments, we only keep the first two categories, which turns the experiments into binary classification problem. The model we used for adversarial training is ResNet20 (He et al., 2015), which is a CNN with 20 layers. The experiments are conducted in $l_2$ and $l_\infty$ space.

Since we can not compute the exact distance between data and classifier, we use the loss on training data [8] with perturbation to represent the robustness of trained model. The perturbations are founded by running 10 steps projected gradient descent (PGD). We handle adversarial training by following the settings in Madry et al. (2018). We use PGD to find the $\arg\max_{\boldsymbol{x}:\|\boldsymbol{x}-\boldsymbol{x}_i\|\leq\varepsilon} \ell(\boldsymbol{x}, \boldsymbol{\theta})$ for each $\boldsymbol{x}_i$.

---

[8]Here we only focus on training data because we do not discuss the generalization of model obtained by adversarial training.

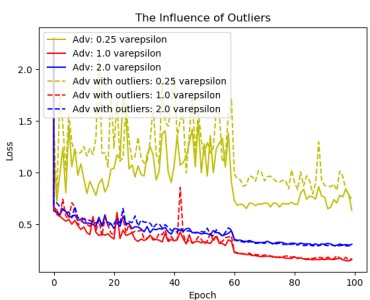 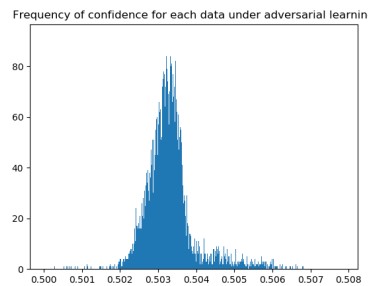

(a) Adversarial training with different $\varepsilon$ for outliers exist or not.

(b) Frequency of confidence for data point under adversarial training with an extremely large $\varepsilon$.

Figure 9: Performance of adversarial training in $l_2$ space

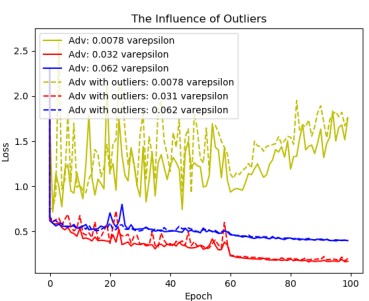 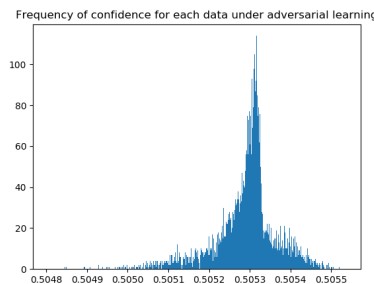

(a) Adversarial training with different $\varepsilon$ for outliers exist or not.

(b) Frequency of confidence for data point under adversarial training with an extremely large $\varepsilon$.

Figure 10: Performance of adversarial training in $l_\infty$ space

The steps of PGD is 10 for each $\boldsymbol{x}_i$. All the models are trained by stochastic gradient descent with 0.1 learning rate and 0.9 momentum parameter for 100 epochs. The loss function is set to be cross entropy.

We first validate our conclusions for $\varepsilon$-strongly linearly separable data. Since neural network is not a linear model, it is hard to construct extra support vectors like in Section 5. Hence, we can only verify our conclusion that iterates of adversarial training converge faster to a robust solution, and it will benefit from an appropriate choice of adversarial radius $\varepsilon$. To compare the influence of $\varepsilon$, we choose $\varepsilon = 0.25, 1.0, 2.0$ in $l_2$ space and $\varepsilon = 2/255, 8/255, 16/255$ in $l_\infty$ space. The size of perturbations are from Madry et al. (2018). Besides that, the perturbations on training set [9] to verify the robustness of trained model are 1.0 and 8/255 respectively in $l_2$ space and $l_\infty$ space. The experimental results in $l_2$ and $l_\infty$ space can be referred to Figure 9a [10] and 10a. The label of curves, for example, "Adv: 0.25 varepsilon" means adversarial training with $\varepsilon = 0.25$. From the figures, we see that the models returned by adversarial training with $\varepsilon = 1$ and $\varepsilon = 8/255$ are respectively the most robust in $l_2$ and $l_\infty$ space. Hence, the conclusion that adversarial training can benefit from an appropriate choice of $\varepsilon$ also holds for neural network.

Then, for non-$\varepsilon$-strongly linearly separable data, we respectively discuss the influence of outliers and a large $\varepsilon$. First, we construct 50 outliers $\{\boldsymbol{x}_i\}_{i=1}^{100} \sim \mathcal{N}(\pm 0.078125 \cdot \mathbf{1}_{32 \times 32 \times 3}, 0.045 \cdot \mathbf{I}_{32 \times 32 \times 3})$ in each category in both $l_2$ and $l_\infty$ space. The influence of outliers for adversarial training in $l_2$ and $l_\infty$ space can be respectively referred to Figure 9a and 10a. "Adv with outliers: 0.25 varepsilon"

---

[9] We should highlight that the perturbations are recalculated after each epoch of training.

[10] We only list the results of adversarial leaning, because the loss on data with perturbation of standard training will not converge (always larger than 2.0).

means adversarial training with outliers added into dataset and $\varepsilon = 0.25$. From the results, we see that outliers will barely effect the performance of adversarial training. Hence, we can conclude that adversarial training is still stable to outliers, even the model is neural network. On the other hand, we respectively choose $\varepsilon$ equal to $4$ and $32/255$ in $l_2$ and $l_\infty$ space to verify the confidence of classifier returned by adversarial training. The distributions of confidence via adversarial training in $l_2$ and $l_\infty$ space can be respectively referred to Figure 9b and 10b. The results reveal that the confidence of neural network trained by adversarial training with large $\varepsilon$ is at a fairly low level in each data.

To summary, although our conclusion about adversarial training are derived from linear classifier, the empirically results suggest it can be somehow extended to the case of neural networks. Thus, a theoretical exploration for neural network is crucial in the future work.

