# OpenReview forum: "THE EFFECT OF ADVERSARIAL TRAINING: A THEORETICAL CHARACTERIZATION"
_ICLR.cc/2020/Conference — Reject_

### Official Review · AnonReviewer3 · 2019-10-16
**Official Blind Review #3**

**Rating:** 1

**Review:**

TL;DR: The paper gives interesting, theoretical results to adversarial training. The paper only uses linear classifiers, which are hardly the same problem as deep networks where adversarial attacks are problematic. Some conclusions from theorems can be vague or informal, and therefore are not very convincing. I vote for rejecting this paper since it is hard to claim it informs deep learning research (the motivating reason for doing adversarial training). However, I am not familiar with theoretical analysis of adversarial attack/defense, so I am open to counter-arguments.

=================
1.    What is the specific question/problem tackled by the paper?
The paper gives a theoretical analysis to the theoretically less-studied procedure of adversarial training, and shows properties of adversarial training in comparison to regular training, for both linearly separable data or inseparable data. The paper sheds light on some empirical behavior of adversarially trained networks, namely that they are more robust to outliers and lower in performance.

2.    Is the approach well motivated, including being well-placed in the literature?
I am not an expert of adversarial samples, so I am ill-equipped to judge the novelty of the paper. The research direction itself is well motivated, and in the realm of deep learning, it is posed as a first paper to theoretically analyze adversarial training.
However, the authors only analyzed linear classifiers. This makes the results of the paper ill-suited for deep networks, whose non-linearity is arguably the reason why adversarial samples are such a problem. The motivation of the paper is thus greatly diminished. For linear classifiers, I do not know if there are existing work on their robustness when perturbation of samples are being trained on, but to be well-placed in the literature, the authors must either claim there is none, or cite those papers.

3.    Does the paper support the claims? This includes determining if results, whether theoretical or empirical, are correct and if they are scientifically rigorous.
Claims and novelties in this paper include:
(1) Adversarial training converges faster than regular training if samples are ε-strongly linearly separable,
(2) If samples are not "ε-strongly linearly separable", adversarial training is robust to outliers, while regular training is not,
(3) Confidence is low for all (training) samples if ε is large.

Only (1) seems to be sufficiently proved. I am not certain that this is a very useful result, and I am open to counter-arguments.
(2) and (3) have steps that are vague and informal:

(2) That regular training is susceptible to outliers is proof by example and people already know that. Also the claim relies on the assumption that pi^1 and pj^2 are on the same scale, while those samples that violate the decision boundary can have arbitrarily large pi^1 or pj^2. Since outliers are often the violators, and inliers often are not, a small number of carefully placed outliers can make ||p2|| quite large while each pi^1 can be very small. It is also worth noting the logic seems to boil down to "N1>>N2, so inliers should overwhelm outliers, making the training robust". The claim is not guaranteed.

(3) I am not very sure if I understand it correctly, but the logic seems to be that the logits are bounded, so they cannot be too large, and so the confidence is low. However, the bound also involves the magnitude of w and x from the other class, so the final step of the proof is either unclear or the bound can indeed be quite large. Note that |wx| does not need to be very large for the confidence to be high (e.g. if logit is 5, the confidence is 1/(1+e^-5)=99.3%). The claim also relies on the assumption that epsilon is larger than the distance between the farthest points in the dataset, which is extreme since you can find an adversarial sample that can be considered to be simultaneously "close enough" to the two most dissimilar samples in the dataset.

In the end, the results are not very convincing or useful for informing deep learning research.

=============
To improve paper:
- Clarify motivation and how this would inform adversarial training highly non-linear classifiers;
- Add related work for robustness to perturbation of linear models, or state that they don't exist;
- Clarify weaknesses in the claims.

Editorial changes:
Definition 2: "logit" <--- people call this probability estimates; logits are wTx
Sec. 4.1.2 needs to clarify what k in x_i^k means -- it's continued from proposition 1 which at first glance is irrelevant


=================
Post rebuttal

Apologies for not interacting earlier due to deadlines.
The rebuttal does not address my major concern (motivation), nor does it discuss its relationship with related work. The math questions are not answered very clearly; I do not see how section 4.1.2 proves p_i^1 has the same scale as p_j^2, except maybe that they are all smaller than some constant for certain samples. And other discussions in the rebuttal simply confirms my concern.
In summary, I think this paper is not yet ready for publication.



**Experience Assessment:**

I do not know much about this area.

**Review Assessment: Checking Correctness Of Derivations And Theory:**

I assessed the sensibility of the derivations and theory.

**Review Assessment: Checking Correctness Of Experiments:**

I did not assess the experiments.

**Review Assessment: Thoroughness In Paper Reading:**

I made a quick assessment of this paper.

---

> ### Author Response · Authors · 2019-11-12
> **Thanks for your review, the following are our responses to your questions**
>
> “For linear classifiers, I do not know if there are existing work on their robustness when perturbation of samples are being trained on, but to be well-placed in the literature, the authors must either claim there is none, or cite those papers”
> To the best of our knowledge, theoretical analysis to the limited point of adversarial training only appeared in paper Ilyas et al., 2019 which we give a discussion between our difference in the last of section 4.1.2. The robustness of limited point is a hard problem, even under linear framework and standard training. Some recent works study it since 2017 i.e. Soudry et al. (2017), Ji & Telgarsky (2018). But this is the first paper to discuss in for adversarial training.
>
> “That regular training is susceptible to outliers is proof by example and people already know that…”
> The standard training is susceptible to outliers is provided in proposition 1 and illustrated by example 1 and example 2. The stability of adversarial training is not build on the assumption “p_{i}^{1} has same scale with p_{j}^{2}”. On the contrary, we show that p_{i}^{1} has same scale with p_{j}^{2} in the first paragraph of section 4.1.2. This is because the limited point of adversarial training w^{*} has a finite norm, otherwise it will end up with a infinite loss due to the non-separability of data.
>
> “I am not very sure if I understand it correctly, but the logic seems to be that the logits are bounded”
> Equation (16) can be viewed as the confidence e^{|w^{T}x_{k,i}|} will be upper bounded by the average loss among the opposite class (smaller than some constant). Our result is e^{|w^{T}x_{k,i}|} can be upper bounded rather than |w^{T}x_{k,i}|.
> The assumption of large \epsilon is a technical assumption in order to give a quantitively description of confidence. A relatively smaller \epsilon can also correspond with low confidence in practice.
>
> Definition 2: "logit" <--- people call this probability estimates; logits are wTx Sec. 4.1.2 needs to clarify what k in x_i^k means -- it's continued from proposition 1 which at first glance is irrelevant.
> 	We will revise the notations accordingly to make this paper more friendly to read in the next version.

---

### Official Review · AnonReviewer2 · 2019-10-23
**Official Blind Review #2**

**Rating:** 1

**Review:**

This paper provides some analyses of the difference between adversarial training and standard training for linear classification problem. In particular, it proves that when the data is \eps linearly separable, adversarial training converges faster than standard trading. It also argues that when the data is not \eps linearly separable, adversarial training is more robust to outlier. Simulations are constructed to verify the arguments in the paper but there is no experiments on real dataset.

The first result of this paper is interesting, that adversarial training converges faster than standard training. Studying the difference between the convergent points between adversarial training and standard training is also an interesting research problem. However, I still have two main concerns about the current version of the paper.
1. The paper is trying to develop rigorous results, but its writing is arguably not rigorous. Many statement are not clear and some notations are used without definition. Section 4.1 has many vague statements. See more concrete comments below.
2. I am not sure about the significance of the results in the paper. The results highly depend on the linear setting with convex losses. More than that, Theorem 1 assumes the \eps strongly linear separable, and Theorem 2 assumes a large \eps (if the statement is that |w* x_{k,i}| is less than a large number, it seems much less interesting). These are very strong assumptions that are usually not true in practice. Experimental results only cover carefully designed simulation as well.

Detailed comments for item 1 above:
1. Assumption 3, what is the quantifier for w? Is it for every w? There exists some w? How do you guarantee by “rescale the norm of w” (from the footnote) to make sure that c_1 is not -\infty?
2. Lemma 1, this is for every x_i, or some particular x_i?
3. What is w(t)?
4. What is the condition on \eta in Theorem 1? Why it is O(\eta) in equation (11) or equation (24)?
5. The claims in section 4.1 seem to be depended on carefully designed examples. Would it still true rigorously for general cases?
6. In section 4.1.2 first paragraph, why ||w_t|| can not go to infinity? in the third paragraph, how assumption 2 implies p^k_i / p^k`_j = o(1), or later  p^k_i / p^k`_j = O(1)?
7. In section 4.2 second paragraph, what is “k_th category”?
8. Is w^* unique in Theorem 2?


**Experience Assessment:**

I have published one or two papers in this area.

**Review Assessment: Checking Correctness Of Derivations And Theory:**

I did not assess the derivations or theory.

**Review Assessment: Checking Correctness Of Experiments:**

I assessed the sensibility of the experiments.

**Review Assessment: Thoroughness In Paper Reading:**

I read the paper at least twice and used my best judgement in assessing the paper.

---

> ### Author Response · Authors · 2019-11-12
> **Thanks for your review, the following are our responses to your questions**
>
> “Simulations are constructed to verify the arguments in the paper but there is no experiments on real dataset.”
> 	A series of experiments on CIFAR10 related to our results are delegated to appendix E.2.
> “Theorem 1 assumes the \eps strongly linear separable, and Theorem 2 assumes a large \eps”
> The assumption “\eps strongly linear separable” is equivalent to linearly separable which implies there must exist a \epsilon>0 such that w^{*T}x_{i} \geq \epsilon \|w^{*}\|. We can accordingly adjust the adversarial radius, then it is a medium assumption. The assumption of large \epsilon in theorem 2 tries to give a quantitively characterization to confidence of classifier obtained by adversarial training. A slightly small \epsilon can also correspond with similar observation in practice. We will try to get rid of this technical assumption in the next version.
>
> “How do you guarantee by “rescale the norm of w” (from the footnote) to make sure that c_1 is not -\infty”
> We should highlight that assumption only appears in theorem 1. The condition is used to ensure L-smoothness of adversarial loss l(w). For every classifier w, we can rescale w as w/\|w\| \max{\|w\|, c} for some small constant c. Then, \|w\| can not equal to zero. Besides that, we can similarly clip the norm of w if w^{T}x_i - \epsilon\|w\| smaller than some negative constant. Then we can ensure c_1 > -\infty. As a matter of fact, we can omit the rescale procedure if the loss function l(u) is \log{1+e^{-u}} or initial learning is smaller than 1/L(w(0)), where L(w(0)) is the local Lipschitz constant of point w(0).
>
> “Lemma 1, this is for every x_i, or some particular x_i”
> The third equation in lemma 1 is for every x_i. It shows that w_t will converge to a point with zero adversarial training loss according to our assumption to the loss function l(u).
>
> “What is w(t). What is the condition on \eta in Theorem 1? Why it is O(\eta) in equation (11) or equation (24)”
> w(t) is defined in equation (6), which is the gradient flow iterates of adversarial training. The behavior of gradient flow iterates can approximate the real iterates obtained by gradient descent w_t here. In theorem 1, we characterize the w(t) to reveal behavior of the real gradient descent iterates w_t. The error bound is given by the learning rate \eta. Hence the O(\eta) in equation (11) and (24) are estimation error between gradient flow iterates w(t) and gradient iterates w_t.
>
> “The claims in section 4.1 seem to be depended on carefully designed examples. Would it still true rigorously for general cases?”
> The two examples are used to illustrate the unstability of standard training, a general description is presented in proposition 1. The stability of adversarial training is delegated to equation (15).
>
> “In section 4.1.2 first paragraph, why \|w_t\| can not go to infinity? in the third paragraph, how assumption 2 implies p^k_i / p^k_j = o(1), or later p^k_i / p^k_j = O(1)?”
> Assumption 2 informs that \lim_{u\to\infty} l(u) = 0, since we assume the data are not separable, then there exists a x_i  such that w^{T}x_i - \epsilon \|w\|\leq 0 for each w. The iterates w_t will converge to a minimum according to lemma 1, while \|w_t\| goes to infinity will rends w^{T}(t)x_i - \epsilon \|w(t)\| goes to -\infinity. Then we will ends up with infinite loss. A more detailed description is referred to paragraph 1 in section 4.1.2.
> Without loss of generality, we use l(u)=e^{-u} as an illustration. p^{k}_{i} = e^{-w^{*T}x_{i}^{k} + \epsilon\|w^{*}\|}. Then p^{k}_{i} / p^{k’}_{j} = \exp{-w^{*T}(x_{i}^{k} – x_{j}^{k})}. The minimum w^{*} has infinite norm which makes the loss goes to zero. Then, p^{k}_{i} / p^{k’}_{j} = o(1), since x_{i}^{k} is non-support vector and x_{j}^{k’} is support vector (w^{*T}(x_{i}^{k} – x_{j}^{k})<0). But for adversarial training \|w^{*}\| is not infinity then p_{i}^{k} will have the same scale for each I and k.
>
> “In section 4.2 second paragraph, what is “k_th category”?”
> 	It represents the data is from the first or the second class.
>
> “Is w^* unique in Theorem 2?”
> 	w^{*} is the hard margin solution, it can be non-unique.

---

> > ### Comment · AnonReviewer2 · 2019-11-15
> > **After Rebuttal**
> >
> > I don't think the response addresses my questions, nor the paper is ready for publication.
> >
> > Just to list a few here. In the response Lemma 1 is used for multiple questions. Note that in Lemma 1 the dataset is assumed to be “\eps strongly linear separable”. Why can it still be used in section 4? Since w* may not be unique, how do I guarantee the convergence of the algorithm? I don't think clipping can be seen as rescaling. Also, will the claims in the paper still hold given the clipping?

---

### Official Review · AnonReviewer1 · 2019-10-29
**Official Blind Review #1**

**Rating:** 1

**Review:**

The aim of this paper is to provide a theoretical analysis of adversarial training under the linear classification setting. The main result states that, under many technical assumptions,  adversarial training using gradient descent may converge to the hard margin SVM classifier with a fast rate. Here "fast" is not the standard 1/T fast rates but, rather, a rate of o(1/log T) (in comparison to recent results that looked into the convergence of gradient descent with logistic loss to the hard-margin SVM solution).

Overall, the paper is not recommended for publication for many reasons.

First, the notation used is sometimes imprecise and in some cases it is entirely wrong. For example, the authors decided to use x_i to replace the product y _i x_i in order to "simplify" notation. This makes things really hard to follow. The authors need to use a different symbol, such as z to stand for the product yx. Second, some equations do not appear to be correctly typed (e.g. in Page 1, standard learning should have xi not x). Third, the authors use epsilon to denote two different things (one for the definition of linearly separable data and one for the robustness radius), and so on.

Second, the paper needs to be proof-read. It has a lot of typos and grammatical errors that make sentences difficult to understand. Examples include:
- "while there several outliers are not or even not linearly separable",
- "a simple generalization error bound informs the high loss on test set"

Third, some of the mathematical results do not make sense. For example,  Definition 1 has two conditions, the first one is immediately satisfied once the second condition is satisfied, so why both? Also, in Proposition 1, the authors should mention that both sets are subsets of a linearly separable superset. In that case, the conclusion of Proposition 1 is obvious. Definitely, if data are linearly separable, then the hard margin will shrink as more examples are added (it cannot increase by definition of "maximum" margin). Moreover, in Example 1, the authors conclude with an inequality of norms and I don't see how this follows from the description of the example. The example is generic and there is nothing that indicates one norm would be larger than the other.

Forth, the authors make many assumptions about the loss without mentioning one example that satisfies them. In fact, the loss function they used later in the experiments violates Assumption 2.

Some additional comments:
- How did the authors arrive at Eq 4? I don't see how this follows from the assumptions. Can you please elaborate?
- I am not aware of any book written by Vapnik in 1995 called "Convex Optimization". I think the authors meant the SVM paper.

Given all of these issues and the fact that the main result is incremental and holds under a very limited setting (homogenous linear classifiers under a strong separability assumption) and relies on very strong assumptions about the loss function that may not be achievable to begin with, I do not recommend acceptance.

===========
#Post Rebuttal Remarks

Regarding the rate of convergence, just to be clear, I was only clarifying what the term "fast" meant in the paper, not complaining. So, this did not affect my score. The notation issues need to be fixed and the authors need to ensure that mathematical equations are written precisely, as stated in my review.

Thanks for clarifying the issue about the loss functions. I agree that the example they mentioned satisfy it but it is not a common loss used in practice.

**Experience Assessment:**

I have read many papers in this area.

**Review Assessment: Checking Correctness Of Derivations And Theory:**

I assessed the sensibility of the derivations and theory.

**Review Assessment: Checking Correctness Of Experiments:**

I did not assess the experiments.

**Review Assessment: Thoroughness In Paper Reading:**

I made a quick assessment of this paper.

---

> ### Author Response · Authors · 2019-11-12
> **Thanks for your review, the following are our responses to your questions**
>
> “Here "fast" is not the standard 1/T fast rates but, rather, a rate of o(1/log T)”
> We try to reveal the actual convergence rate of adversarial training to robust solution rather than developing a faster algorithm to obtain a robust solution. Hence, we think the result is valuable, even though the promotion is not surprising large to O(1/T).
> “the authors decided to use x_i to replace the product y _i x_i in order to "simplify" notation.”
> We would like to clarify the notations here and accordingly revise them in the paper. First, representing y_i x_i by x_i is out of the consideration that x_i and y_i are always appear as x_i y_i and w^{T}x_i y_i means the data is correctly classified. The symbol will not mislead the notations. We will substitute the \epsilon in definition 2 with some other symbol.
> “Third, some of the mathematical results do not make sense”
> The two conditions in definition 1 are used to emphasis that w^{*} can not only make a correct classification for each data but also ensure each data is away from the margin larger than \epsilon. We will get rid of the first condition in the next version. The condition that union data is linearly separable is necessary to make proposition meaningful. We will then add it. The example 1 is used to give an intuitively explanation to proposition 1. It can be concluded as the hard margin solution can be sensitive to outliers closely with each other, but standard training can converge to hard margin solution which informs the unstability of standard training. The equation (12) is a direct induction to reveal the x_{i}^{1} will locate closely to the new hard margin solution (\hat{w}_{1}, \hat{w}_{2}) due to some outliers \x_{i}^{2} are added, while they locate away from their original hard margin solution (\hat{\w}_{1}, 0).
> “Forth, the authors make many assumptions about the loss without mentioning one example that satisfies them”
> We enumerate some loss functions satisfy our assumptions in the footnote of page 3. For example, choosing loss function l(u) as e^{-u} or \log{1+e^{-u}} can satisfy with our assumptions. Also please notice that assumption 2 exponential tail focuses on the behavior of loss function l(u) when u is large. We chose l(u)=\log{1+e^{-u}} which will closely to e^{-u} when u is large. A simple inequality that \log{1+x} \geq x – x^{2}/2 can give the exact conclusion.
> “How did the authors arrive at Eq 4”
> It is a directly result according to the linear classifier and monotonically decreasing property of loss function l(u).
>
> “under a very limited setting (homogenous linear classifiers under a strong separability assumption)”
> Our assumptions about loss function can be achieved by the most generally used loss functions i.e. e^{-u}, \log{1+e^{-u}}. Also about the assumption “strongly separability”, it is equivalent to the data are linearly separable, because if the data are separable, then there must exist a \epsilon>0 such that w^{*T}x_{i} \geq \epsilon\|\w^{*}\|. Then the data can satisfy our “\epsilon-strongly linearly separable”. Hence, our core assumption is the data are themselves linearly separable. Besides that, we also give a discussion to adversarial training on non-linearly separable data.

---

### Decision · Program_Chairs · 2019-12-19

**Decision:**

Reject

**Comment:**

This paper studies adversarial training in the linear classification setting, and shows a rate of convergence for adversarial training of o(1/log T) to the hard margin SVM solution under a set of assumptions.

While 2 reviewers agree that the problem and the central result is somewhat interesting (though R3 is uncertain of the applicability to deep learning, I agree that useful insights can often be gleaned from studying the linear case), reviewers were critical of the degree of clarity and rigour in the writing, including notation, symbol reuse, repetitions/redundancies, and clarity surrounding the assumptions made.

No updates to the paper were made and reviewers did not feel their concerns were addressed by the rebuttals. I therefore recommend rejection, but would encourage the authors to continue refining their paper in order to showcase their results more clearly and didactically.